# RELAY DIFFUSION: UNIFYING DIFFUSION PROCESS ACROSS RESOLUTIONS FOR IMAGE SYNTHESIS

**Jiayan Teng**[⋆1]**, Wendi Zheng**[⋆1]**, Ming Ding**[⋆12†]**,**
**Wenyi Hong**[1]**, Jianqiao Wangni**[2]**, Zhuoyi Yang**[1]**, Jie Tang**[1†]
[⋆]equal contribution [1]Tsinghua University [2]Zhipu AI [†] corresponding authors
{tengjy20@mails,zhengwd23@mails,jietang@mail}.tsinghua.edu.cn
mingding.thu@gmail.com

## ABSTRACT

Diffusion models achieved great success in image synthesis, but still face challenges in high-resolution generation. Through the lens of discrete cosine transformation, we find the main reason is that *the same noise level on a higher resolution results in a higher Signal-to-Noise Ratio in the frequency domain*. In this work, we present Relay Diffusion Model (RDM), which transfers a low-resolution image or noise into an equivalent high-resolution one for diffusion model via blurring diffusion and block noise. Therefore, the diffusion process can continue seamlessly in any new resolution or model without restarting from pure noise or low-resolution conditioning. RDM achieves state-of-the-art FID on CelebA-HQ and sFID on ImageNet 256×256, surpassing previous works such as ADM, LDM and DiT by a large margin. All the codes and checkpoints are open-sourced at https://github.com/THUDM/RelayDiffusion.

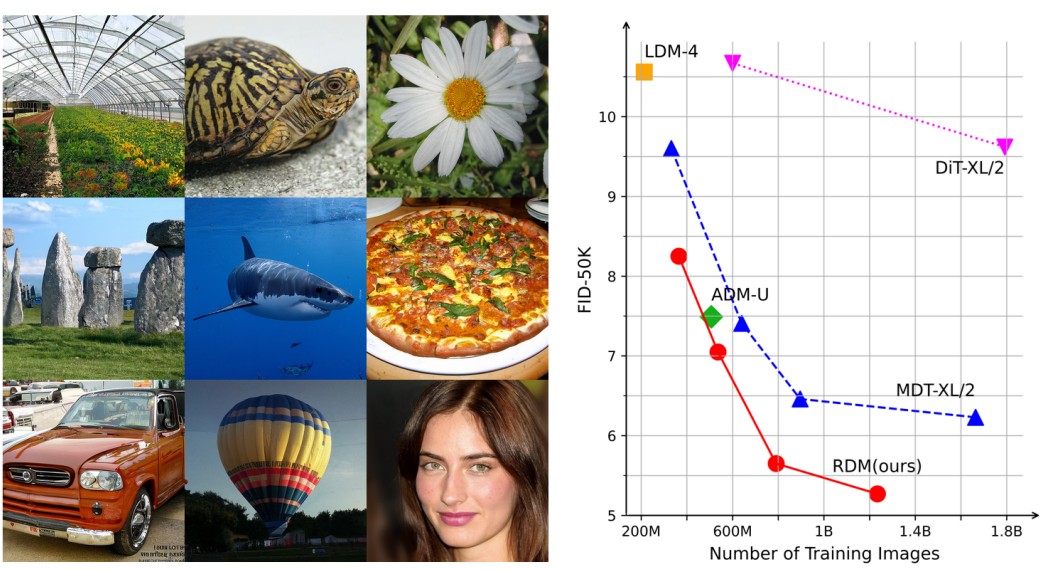

Figure 1: (left): Generated Samples by RDM on ImageNet 256×256 and CelebA-HQ 256×256. (right): Benchmarking recent diffusion models on class-conditional ImageNet 256×256 generation without any guidance. RDM can achieve a FID of 1.99 (and a class-balanced FID of 1.87) if with classifier-free guidance.

## 1 INTRODUCTION

Diffusion models (Ho et al., 2020; Rombach et al., 2022) succeeded GANs (Goodfellow et al., 2020) and autoregressive models (Ramesh et al., 2021; Ding et al., 2021) to become the most prevalent

generative models in recent years. However, challenges still exist in the training of diffusion models for high-resolution images. More specifically, there are two main obstacles:

**Training Efficiency.** Although equipped with UNet to balance the memory and computation cost across different resolutions, diffusion models still require a large amount of resources to train on high-resolution images. One popular solution is to train the diffusion model on a latent (usually $4\times$ compression rate in resolution) space and map the result back as pixels (Rombach et al., 2022), which is fast but inevitably suffers from some low-level artifacts. The cascaded method (Ho et al., 2022; Saharia et al., 2022) trains a series of varying-size super-resolution diffusion models, which is effective but needs a complete sampling for each stage separately.

**Noise Schedule.** Diffusion models need a noise schedule to control the amount of the isotropic Gaussian noise at each step. The setting of the noise schedule shows great influence over the performance, and most current models follow the linear (Ho et al., 2020) or cosine (Nichol & Dhariwal, 2021) schedule. However, *an ideal noise schedule should be resolution-dependent* (See Figure 2 or Chen (2023)), resulting in suboptimal performance to train high-resolution models directly with common schedules designed for resolutions of $32\times32$ or $64\times64$ pixels.

These obstacles hindered previous researchers from establishing an effective end-to-end diffusion model for high-resolution image generation. Dhariwal & Nichol (2021) attempted to directly train a $256\times256$ ADM but found that it performs much worse than the cascaded pipeline. Chen (2023) and Hoogeboom et al. (2023) carefully adjusted the hyperparameters of the noise schedule and architecture for high-resolution cases, but the quality is still not comparable to the state-of-the-art cascaded method (Saharia et al., 2022).

In our opinion, the cascaded method contributes in both training efficiency and noise schedule: (1) It provides flexibility to adjust the model size and architecture for each stage to find the most efficient combination. (2) The existence of low-resolution condition makes the early sampling steps easy, so that the common noise schedules (optimized for low-resolution models) can be applied as a feasible baseline to the super-resolution models. Moreover, (3) high-resolution images are more difficult to obtain on the Internet than low-resolution images. The cascaded method leverages the knowledge from low-resolution samples, meanwhile keeps the capability to generate high-resolution images. Therefore, it might not be a promising direction to completely replace the cascaded method with an end-to-end one at the current stage.

The disadvantages of the cascaded method are also obvious: (1) Although the low-resolution part is determined, a complete diffusion model starting from pure noise is still trained and sampled for super-resolution, which is time-consuming. (2) The distribution mismatch between ground-truth and the generated low-resolution condition will hurt the performance, so that tricks like conditioning augmentation (Ho et al., 2022) become vitally important to mitigate the gap. Besides, the noise schedule of high-resolution stages are still not well studied.

**Present Work.** Here we present the **R**elay **D**iffusion **M**odel (RDM), a new cascaded framework to improve the shortcomings of the previous cascaded methods. In each stage, the model starts diffusion from the result of the last stage, instead of conditioning on it and starting from pure noise. Our method is named as the cascaded models work together like a "relay race". The contributions of this paper can be summarized as follows:

- We analyze the reasons of the difficulty of noise scheduling in high-resolution diffusion models in frequency domain. Previous works like LDM (Rombach et al., 2022) assume all image signals from the same distribution when analyzing the SNR, neglecting the difference in frequency domain between low-resolution and high-resolution images. Our analysis successfully accounts for phenomenon that the same noise level shows different perceptual effects on different resolutions, and introduce the *block noise* to bridge the gap.

- We propose RDM to disentangle the diffusion process and the underlying neural networks in the cascaded pipeline. RDM gets rid of the low-resolution conditioning and its distribution mismatch problem. Since RDM starts diffusion from the low-resolution result instead of pure noise, the training and sampling steps can also be reduced.

- We evaluate the effectiveness of RDM on unconditional CelebA-HQ $256\times256$ and conditional ImageNet $256\times256$ datasets. RDM achieves state-of-the-art FID on CelebA-HQ and sFID on ImageNet.

## 2 PRELIMINARY

### 2.1 DIFFUSION MODELS

To model the data distribution $p_{data}(\mathbf{x}_0)$, denoising diffusion probabilistic models (DDPMs, Ho et al. (2020)) define the generation process as a Markov chain of learned Gaussian transitions. DDPMs first assume a forward diffusion process, corrupting real data $\mathbf{x}_0$ by progressively adding Gaussian noise from time steps 0 to $T$, whose variance $\{\beta_t\}$ is called the noise schedule:

$$q(\mathbf{x}_t|\mathbf{x}_{t-1}) = \mathcal{N}(\mathbf{x}_t; \sqrt{1 - \beta_t}\mathbf{x}_{t-1}, \beta_t\mathbf{I}). \tag{1}$$

The reverse diffusion process is learned by a time-dependent neural network to predict denoised results at each time step, by optimizing the variational lower bound (ELBO).

Many other formulations for diffusion models include stochastic differential equations (SDE, Song et al. (2020b)), denoising diffusion implicit models (DDIM, Song et al. (2020a)), etc. Karras et al. (2022) summarizes these different formulations into the **EDM** framework. In this paper, we generally follow the EDM formulation and implementation. The training objective of EDM is defined as $L_2$ error terms:

$$\mathbb{E}_{\mathbf{x}\sim p_{data}, \sigma\sim p(\sigma)}\mathbb{E}_{\epsilon\sim\mathcal{N}(\mathbf{0},\mathbf{I})}\|D(\mathbf{x} + \sigma\epsilon, \sigma) - \mathbf{x}\|^2, \tag{2}$$

where $p(\sigma)$ represents the distribution of a continuous noise schedule. $D(\mathbf{x} + \sigma\epsilon, \sigma)$ represents the denoiser function depending on the noise scale. We also follow the EDM precondition for $D(\mathbf{x} + \sigma\epsilon, \sigma)$ with $\sigma$-dependent skip connection (Karras et al., 2022).

Cascaded diffusion model (CDM, Ho et al. (2022)) is proposed for high-resolution generation. CDM divides the generation into multiple stages, where the first stage generates low-resolution images and the following stages perform super-resolution conditioning on the outputs of the previous stage. f-DM (Gu et al., 2022) unifies multiple resolutions of image generation with a linear interpolation in a single model. Cascaded models are extensively adopted in recent works of text-to-image generation, e.g. Imagen (Saharia et al., 2022), DALL-E-2 (Ramesh et al., 2022) and eDiff-I (Balaji et al., 2022).

### 2.2 BLURRING DIFFUSION

The Inverse Heat Dissipation Model (IHDM) (Rissanen et al., 2022) generates images by reversing the heat dissipation process. The heat dissipation is a thermodynamic process describing how the temperature $u(x, y, t)$ at location $(x, y)$ changes in a (2D) space with respect to the time $t$. The dynamics can be denoted by a PDE $\frac{\partial u}{\partial t} = \frac{\partial^2 u}{\partial x^2} + \frac{\partial^2 u}{\partial y^2}$.

Blurring diffusion (Hoogeboom & Salimans, 2022) is further derived by augmenting the Gaussian noise with heat dissipation for image corruption. Since simulating the heat equation up to time $t$ is equivalent to a convolution with a Gaussian kernel with variance $\sigma^2 = 2t$ in an infinite plane (Bredies et al., 2018), the intermediate states $\boldsymbol{x}_t$ become blurry, instead of noisy in the standard diffusion. If Neumann boundary conditions are assumed, blurring diffusion in discrete 2D pixel space can be transformed to the frequency space by Discrete Cosine Transformation (DCT) conveniently as:

$$q(\boldsymbol{u}_t|\boldsymbol{u}_0) = \mathcal{N}(\boldsymbol{u}_t|\boldsymbol{D}_t\boldsymbol{u}_0, \sigma_t^2\boldsymbol{I}), \tag{3}$$

where $\boldsymbol{u}_t = \text{DCT}(\boldsymbol{x}_t)$, and $\boldsymbol{D}_t = e^{\boldsymbol{\Lambda}t}$ is a diagonal matrix with $\boldsymbol{\Lambda}_{i\times W+j} = -\pi^2(\frac{i^2}{H^2} + \frac{j^2}{W^2})$ for coordinate $(i, j)$. Here Gaussian noise with variance $\sigma_t^2$ is mixed into the blurring diffusion process to transform the deterministic dissipation process to a stochastic one for diverse generation.

## 3 METHOD

### 3.1 MOTIVATION

The noise schedule is vitally important to the diffusion models and is resolution-dependent. A certain noise level appropriately corrupting the $64 \times 64$ images, could fail to corrupt the $256 \times 256$ (or a higher resolution) images, which is shown in the first row of Figure 2(a)(b). Chen (2023) and Hoogeboom et al. (2023) attributed this to the lack of schedule-tuning, but we found that an analysis from the perspective of frequency spectrum can help us better understand this phenomenon.

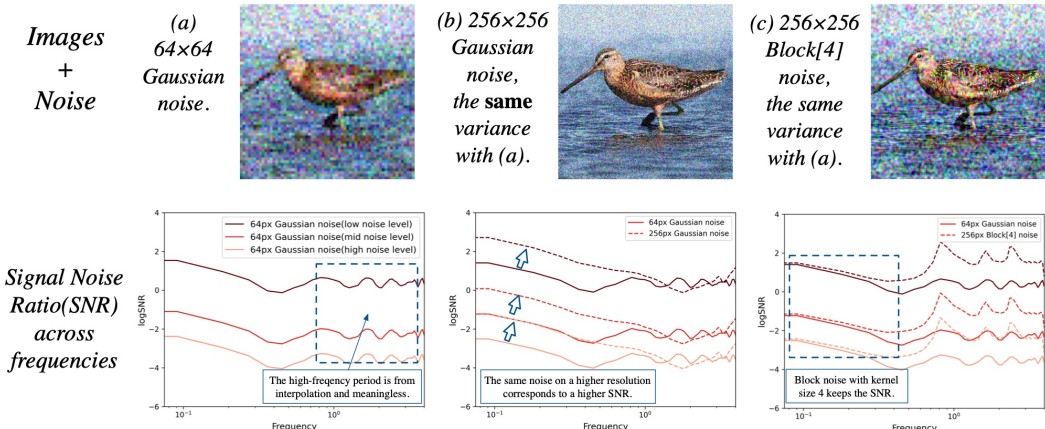

Figure 2: Illustration of spatial and frequency results after adding independent Gaussian and block noise. (a)(b) At the resolution of $64 \times 64$ and $256 \times 256$, the same noise level results in different perceptual effects, and in the frequency plot, the SNR curve shifts upward. (c) The independent Gaussian noise at the resolution $64 \times 64$ and block noise (kernel size = 4) at the resolution $256 \times 256$ produce similar results in both spatial domain and frequency domain. The noise is $\mathcal{N}(0, 0.3^2)$ for (a). These SNR curves are universally applicable to most natural images.

**Frequency spectrum analysis of the diffusion process.** The natural images with different resolutions can be viewed as the result of visual signals sampled at varying frequencies. To compare the frequency features of a $64 \times 64$ image and a $256 \times 256$ image, we can upsample the $64 \times 64$ one to $256 \times 256$, perform DCT and compare them in the 256-point DCT spectrum. The second row of Figure 2(a) shows the signal noise ratio (SNR) at different frequencies and diffusion steps. In Figure 2(b), we clearly find that *the same noise level on a higher resolution results in a higher SNR in (the low-frequency part of) the frequency domain*. Detailed frequency spectrum analysis are included in Appendix D.

At a certain diffusion step, a higher SNR means that during training the neural network presumes the input image more accurate, but the early steps may not be able to generate such accurate images after the increase in SNR. This training-inference mismatch will accumulate over step by step during sampling, leading to the degradation of performance.

**Block noise as the equivalence at high resolution.** After the upsampling from $64 \times 64$ to $256 \times 256$, the independent Gaussian noise on $64 \times 64$ becomes noise on $4 \times 4$ grids, thus greatly changes its frequency representation. To find a variant of the $s \times s$-grid noise without deterministic boundaries, we propose **Block noise**, where the Gaussian noises are correlated for nearby positions. More specifically, the covariance between noise $\epsilon_{x_0,y_0}$ and $\epsilon_{x_1,y_1}$ is defined as

$$\mathrm{Cov}(\epsilon_{x_0,y_0}, \epsilon_{x_1,y_1}) = \frac{\sigma^2}{s^2} \max\left(0, s - \mathrm{dis}(x_0, x_1)\right) \max\left(0, s - \mathrm{dis}(y_0, y_1)\right), \quad (4)$$

where $\sigma^2$ is the noise variance, and $s$ is a hyperparameter *kernel size*. The $\mathrm{dis}(\cdot, \cdot)$ function here is the Manhattan distance. For simplicity, we "connect" the top and bottom edges and the left and right edges of the image, resulting in

$$\mathrm{dis}(x_0, x_1) = \min\left(|x_0 - x_1|, x_{max} - |x_0 - x_1|\right). \quad (5)$$

The block noise with kernel size $s$ can be generated by averaging $s \times s$ independent Gaussian noise. Suppose we have an independent Gaussian noise matrix $\epsilon$, the block noise construction function $\mathrm{Block}[s](\cdot)$ is defined as

$$\mathrm{Block}[s](\epsilon)_{x,y} = \frac{1}{s} \sum_{i=0}^{s-1} \sum_{j=0}^{s-1} \epsilon_{x-i,y-j}, \quad (6)$$

where $\mathrm{Block}[s](\epsilon)_{x,y}$ is the block noise at the position $(x, y)$, and $\epsilon_{-x} = \epsilon_{x_{max}-x}$. Figure 2(c) shows that the block noise with kernel size $s = 4$ on $256 \times 256$ has a similar frequency spectrum as the independent Gaussian noise on $64 \times 64$ images.

The analysis above seems to indicate that we can design an end-to-end model for high-resolution images by introducing block noise in early diffusion steps, while cascaded models already achieves great success. Therefore, a revisit of the cascaded models is necessary.

**Why does the cascaded models alleviate this issue?** Experiments in previous works (Nichol & Dhariwal, 2021; Dhariwal & Nichol, 2021) have already shown that cascaded models perform better than end-to-end models under a fair setting. These models usually use the same noise schedule in all stages, so why are the cascaded models not affected by the increase of SNR? The reason is that in the super-resolution stages, the low-resolution condition greatly eases the difficulty of the early steps, so that although the higher SNR requires a more accurate input, the accuracy is within the capability of the model.

A natural idea is that since the low-frequency information in the high-resolution stage has already been determined by the low-resolution condition, we can continue generating directly from the up-sampled result to reduce both the training and sampling steps. However, the generation of low-resolution images is not perfect, and thus the solution of the distribution mismatch between ground-truth and generated low-resolution images is a priority to "continue" the diffusion process.

## 3.2 RELAY DIFFUSION

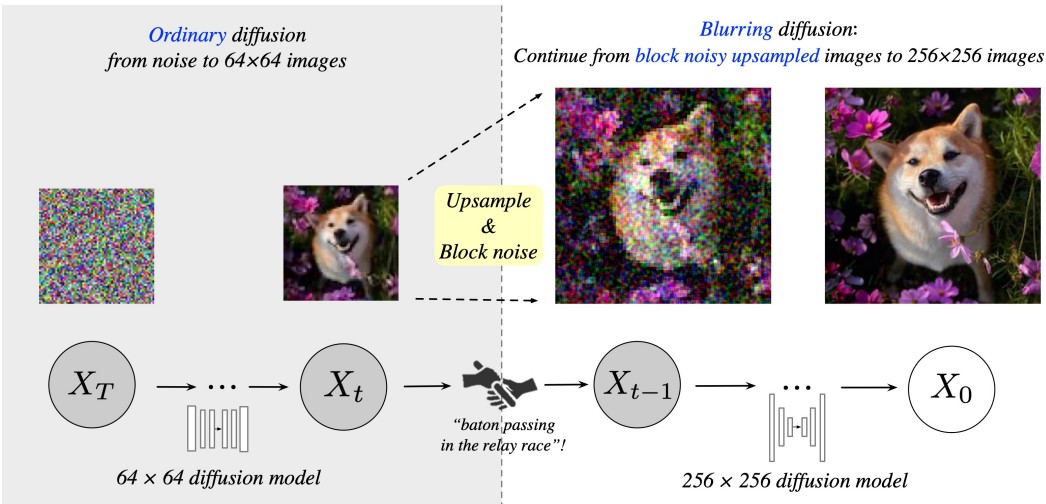

Figure 3: Pipeline of Relay Diffusion Models (RDM).

We propose relay diffusion model (RDM), a cascaded pipeline connecting the stages with block noise and (patch-level) blurring diffusion. Different from CDM, RDM considers the equivalence of the low-resolution generated images when upsampled to high resolution. Suppose that the generated $64 \times 64$ low-resolution image $\mathbf{x}_0^L = \mathbf{x}^L + \epsilon_L$ can be decomposed into a sample in real distribution $\mathbf{x}^L$ and a remaining noise $\epsilon_L \sim \mathcal{N}(\mathbf{0}, \beta_0^2 \mathbf{I})$. As mentioned in section 3.1, the $256 \times 256$ equivalence of $\epsilon_L$ is Block[4] noise with variance $\beta_0^2$, denoted by $\epsilon_H$. After (nearest) upsampling, $\mathbf{x}^L$ becomes $\mathbf{x}^H$, where each $4 \times 4$ grid share the same pixel values. We can define it as the starting state of a *patch-wise blurring diffusion*.

Unlike blurring diffusion models (Rissanen et al., 2022) (Hoogeboom & Salimans, 2022) that perform the heat dissipation on the entire space of images, we propose to implement the heat dissipation on each $4 \times 4$ patch independently, which is of the same size as the upsampling scale. We first define a series of patch-wise blurring matrices $\{\boldsymbol{D}_t^p\}$, which is introduced in detail in Appendix A.1. The forward process would have a similar representation with equation 3:

$$q(\boldsymbol{x}_t|\boldsymbol{x}_0) = \mathcal{N}(\boldsymbol{x}_t|\boldsymbol{V}\boldsymbol{D}_t^p\boldsymbol{V}^{\mathrm{T}}\boldsymbol{x}_0, \sigma_t^{2}\boldsymbol{I}), \quad t \in \{0,..,T\}, \tag{7}$$

where $\boldsymbol{V}^{\mathrm{T}}$ is the projection matrix of DCT and $\sigma_t$ is the variance of noise. Here the $\boldsymbol{D}_t^p$ is chosen to guarantee $\boldsymbol{V}\boldsymbol{D}_T^p\boldsymbol{V}^{\mathrm{T}}\boldsymbol{x}_0$ in the same distribution as $\boldsymbol{x}^H$, meaning that the blurring process ultimately makes the pixel value in each $4 \times 4$ patch the same.

The training objective of the high-resolution stage of RDM generally follows EDM (Karras et al., 2022) framework in our implementation. We replace the Gaussian noise in equation 7 with a mixture of Gaussian noise and block noise in section 3.1. The loss function is defined on the prediction of denoiser function $D$ to fit with true data $\boldsymbol{x}$, which is written as:

$$\mathbb{E}_{\boldsymbol{x}\sim p_{data},t\sim\mathcal{U}(0,1),\epsilon\sim\mathcal{N}(\mathbf{0},\mathbf{I}),\epsilon'\sim\mathcal{N}(\mathbf{0},\mathbf{I})}\|D(\boldsymbol{x}_t,\sigma_t)-\boldsymbol{x}\|^2,$$

$$\text{where}\quad \boldsymbol{x}_t = \underbrace{\boldsymbol{V}\boldsymbol{D}_t^p\boldsymbol{V}^{\mathrm{T}}\boldsymbol{x}}_{blurring} + \frac{\sigma}{\sqrt{1+\alpha^2}}\big(\epsilon+\alpha\cdot\underbrace{\text{Block}[s](\epsilon')}_{block\ noise}\big), \tag{8}$$

where $\epsilon$ and $\epsilon'$ are two independent Gaussian noise. The main difference in training between RDM and EDM is that the corrupted sample $\boldsymbol{x}_t$ is not simply $\boldsymbol{x}_t = \boldsymbol{x} + \epsilon$, but a mixture of the blurred image, block noise and independent Gaussian noise. Ideally, the noise should gradually transfer from block noise to high-resolution independent Gaussian noise, but we find that a weighting average strategy perform well enough, because the low-frequency component of the block noise is much larger than the independent Gaussian noise, and vice versa for high-frequency component. $\alpha$ is a hyperparameter and the normalizer $\frac{1}{\sqrt{1+\alpha^2}}$ is used to keep the variance of the noise, $\sigma^2$ unchanged.

The advantages of RDM compared to CDM includes:

- RDM is more efficient, because RDM skips the re-generation of low-frequency information in the high-resolution stages, and reduce the number of training and sampling steps.

- RDM is simpler, because it gets rid of the low-resolution conditioning and conditioning augmentation tricks. The consumption from cross-attention with the low-resolution condition is also spared.

- RDM is more potential in performance, because RDM is a Markovian denoising process (if with DDPM sampler). Artifacts in the low-resolution images could be corrected in the high-resolution stage, while CDM is trained to correspond to the low-resolution condition.

Compared to end-to-end models (Chen, 2023; Hoogeboom et al., 2023),

- RDM is more flexible to adjust the model size and leverage more low-resolution data.

### 3.3 STOCHASTIC SAMPLER

Since RDM differs from traditional diffusion models in the forward process, we also need to adapt the sampling algorithms. In this section, we focus on the EDM sampler (Karras et al., 2022) due to its flexibility to switch between the first and second order (Heun's) samplers.

Heun's method introduces an additional step for the correction of the first-order sampling. The updating direction of a first-order sampling step is controlled by the gradient term $\boldsymbol{d}_n = \frac{\boldsymbol{x}_n - \boldsymbol{x}_\theta(\boldsymbol{x}_n,\sigma_{t_n})}{\sigma_{t_n}}$. The correction step updates current states with an averaged gradient term $\frac{\boldsymbol{d}_n+\boldsymbol{d}_{n-1}}{2}$. Heun's method takes account of the change of gradient term $\frac{d\boldsymbol{x}}{dt}$ between $t_n$ and $t_{n-1}$. Therefore, it achieves higher quality while allowing for fewer steps of sampling.

We adapt the EDM sampler to the blurring diffusion of RDM's super-resolution stage following the derivation of DDIM (Song et al., 2020a). We define the indices of sampling steps as $\{t_i\}_{i=0}^N$, in corresponding to the noisy states of images $\{\boldsymbol{x}_i\}_{i=0}^N$. To apply blurring diffusion, images are transformed into frequency space by DCT as $\boldsymbol{u}_i = \boldsymbol{V}^{\mathrm{T}}\boldsymbol{x}_i$. Song et al. (2020a) uses a family of inference distributions to describe the diffusion process. We can write it for blurring diffusion as:

$$q_\delta(\boldsymbol{u}_{1:N}|\boldsymbol{u}_0) = q_\delta(\boldsymbol{u}_N|\boldsymbol{u}_0)\prod_{n=2}^N q_\delta(\boldsymbol{u}_{n-1}|\boldsymbol{u}_n,\boldsymbol{u}_0), \tag{9}$$

where $\delta \in \mathbb{R}_{\geq 0}^N$ denotes the index vector for the distribution. For all $n > 1$, the backward process is:

$$q_\delta(\boldsymbol{u}_{n-1}|\boldsymbol{u}_n,\boldsymbol{u}_0) = \mathcal{N}\big(\boldsymbol{u}_{n-1}|\frac{1}{\sigma_{t_n}}(\sqrt{\sigma_{t_{n-1}}^2-\delta_n^2}\boldsymbol{u}_n + (\sigma_{t_n}\boldsymbol{D}_{t_{n-1}}^p - \sqrt{\sigma_{t_{n-1}}^2-\delta_n^2}\boldsymbol{D}_{t_n}^p)\boldsymbol{u}_0), \delta_n^2\boldsymbol{I}\big). \tag{10}$$

The mean of the normal distribution ensures the forward process to be consistent with the formulation of blurring diffusion in Section 3.2, which is $q(\boldsymbol{u}_n|\boldsymbol{u}_0) = \mathcal{N}(\boldsymbol{u}_n|\boldsymbol{D}_{t_n}^p \boldsymbol{u}_0, \sigma_{t_n}^2 \boldsymbol{I})$. We provide a detailed proof of the consistency between our sampler and the formulation of blurring diffusion in Appendix A.3. When the index vector $\delta$ is 0, the sampler degenerates into an ODE sampler. We set $\delta_n = \eta \sigma_{t_{n-1}}$ for our sampler, where $\eta \in [0, 1)$ is a fixed scalar controlling the scale of randomness injected during sampling. We substitute the definition into equation 10 to obtain our sampler function as:

$$\boldsymbol{u}_{n-1} = (\boldsymbol{D}_{t_{n-1}}^p + \gamma_n(\boldsymbol{I} - \boldsymbol{D}_{t_n}^p))\boldsymbol{u}_n + \sigma_{t_n}(\gamma_n \boldsymbol{D}_{t_n}^p - \boldsymbol{D}_{t_{n-1}}^p)\frac{\boldsymbol{u}_n - \tilde{\boldsymbol{u}}_0}{\sigma_{t_n}} + \eta \sigma_{t_{n-1}} \boldsymbol{\epsilon}, \qquad (11)$$

where $\gamma_n \triangleq \sqrt{1 - \eta^2}\frac{\sigma_{t_{n-1}}}{\sigma_{t_n}}$. As in the section 3.1, we also need to consider block noise besides blurring diffusion. The adaptation is just to replace isotropic Gaussian noise $\boldsymbol{\epsilon}$ with $\tilde{\boldsymbol{\epsilon}}$, which is a weighted sum of the block noise and isotropic Gaussian noise. $\tilde{\boldsymbol{u}}_0 = \boldsymbol{u}_\theta(\boldsymbol{u}_n, \sigma_{t_n})$ is predicted by the neural network.

Finally, a stochastic sampler for the super-resolution stage of RDM is summaried in Appendix A.4.

## 4 EXPERIMENTS

### 4.1 EXPERIMENTAL SETTING

**Dataset.** We use CelebA-HQ and ImageNet in our experiments. CelebA-HQ (Karras et al., 2018) is a high-quality subset of CelebA (Liu et al., 2015) which consists of 30,000 images of faces from human celebrities. ImageNet (Deng et al., 2009) contains 1,281,167 images spanning 1000 classes and is a widely-used dataset for generation and other vision tasks. We train RDM on these datasets to generate $256 \times 256$ images. See Appendix C.1 for further experiments on higher resolutions.

**Architecture and Training.** RDM adopts UNet (Ronneberger et al., 2015) as the backbone of diffusion models for all stages. The detailed architectures largely follow ADM (Dhariwal & Nichol, 2021) for fair comparison. We train unconditional models on CelebA-HQ and class-conditional models on ImageNet respectively. Since we follow the EDM implementation, we directly use the released checkpoint from EDM in ImageNet in the $64 \times 64$ stage. We calculate the training consumption by the number of training samples at $256 \times 256$ resolution, while also including the training cost of the $64 \times 64$ stage in the total calculation. According to Appendix B.1, the FLOPs of the $64 \times 64$ model are less than $1/10$ that of the $256 \times 256$ model. So we add $1/10$ of the first stage's number of training samples to the $256 \times 256$ stage's to be the total training consumption. See Appendix B.1 for more information about the architecture and hyperparameters.

**Evaluation.** We use metrics including FID (Heusel et al., 2017), sFID (Nash et al., 2021), IS (Salimans et al., 2016), Precision and Recall (Kynkäänniemi et al., 2019) for a comprehensive evaluation of the results. FID measures the difference between the features of model generations and real images, which is extracted by a pretrained Inception network. sFID differs from FID by using intermediate features, which better measures the similarity of spatial distribution. IS and Precision both measure the fidelity of the samples, while Recall indicates the diversity. We compute metrics with 50,000 and 30,000 generated samples for ImageNet and CelebA-HQ respectively.

Table 1: Benchmarking unconditional image generation on CelebA-HQ $256 \times 256$.

| Unconditional CelebA-HQ $256 \times 256$ | | | | |
|---|---|---|---|---|
| Model | FID↓ | Precision↑ | Recall↑ | Cost(Iter×BS) |
| LSGM (Vahdat et al., 2021) | 7.22 | - | - | 470K×128 |
| WaveDiff (Phung et al., 2023) | 5.94 | - | 0.37 | 234k×64 |
| LDM-4 (Rombach et al., 2022) | 5.11 | 0.72 | 0.49 | 410k×48 |
| StyleSwin (Zhang et al., 2022) | 3.25 | - | - | 25600k×32 |
| **RDM** | **3.15** | **0.77** | **0.55** | 46k×1024 |

Table 2: Effect of stochasticity in the sampler on ImageNet $256 \times 256$ (top) and CelebA-HQ $256 \times 256$ (bottom). We explored different values of $\eta$ in Eq. 11.

| $\eta$ | 0 | 0.10 | 0.15 | 0.20 | 0.25 | 0.30 | 0.40 | 0.50 |
|---|---|---|---|---|---|---|---|---|
| FID↓ | 5.65 | 5.44 | 5.31 | **5.27** | 5.48 | 5.91 | 6.91 | 9.17 |

| $\eta$ | 0 | 0.10 | 0.15 | 0.20 | 0.25 | 0.30 | 0.40 | 0.50 |
|---|---|---|---|---|---|---|---|---|
| FID↓ | 4.11 | 3.74 | 3.43 | **3.15** | 3.23 | 3.52 | 4.79 | 6.41 |

### 4.2 RESULTS

**CelebA-HQ** We compare RDM with the existing methods on CelebA-HQ $256 \times 256$ in Table 1, $512 \times 512$ in Table 6 and $1024 \times 1024$ in Table 7. RDM outperforms the state-of-the-art model

---

[1]*class-balance* means making the number of images generated for each class same among 50,000 images.

Table 3: **Benchmarking class-conditional image generation on ImageNet $256 \times 256$.** The cost of RDM in the table has taken the first-stage model into consideration and made equivalent conversions according to Section 4.1. The cost of latent diffusion model's vae is not taken into consideration. The calculation process of NFE is clarified in sampling steps part of Section 4.3.

| **Class-Conditional ImageNet** $256 \times 256$ | | | | | | | |
|---|---|---|---|---|---|---|---|
| Model | FID↓ | sFID↓ | IS↑ | Precision↑ | Recall↑ | Cost(Iter×BS) | Sampling NFE |
| BigGAN-deep (Brock et al., 2018) | 6.95 | 7.36 | 171.4 | 0.87 | 0.28 | 165k×2048 | - |
| StyleGAN-XL (Sauer et al., 2022) | 2.30 | 4.02 | 265.12 | 0.78 | 0.53 | - | - |
| ADM (Dhariwal & Nichol, 2021) | 10.94 | 6.02 | 100.98 | 0.69 | 0.63 | 1980k×256 | 250 |
| LDM-4 (Rombach et al., 2022) | 10.56 | - | 103.49 | 0.71 | 0.62 | 178k×1200 | 250 |
| CDM (Ho et al., 2022) | 4.88 | - | 158.71 | - | - | - | 100 |
| DiT-XL/2 (Peebles & Xie, 2022) | 9.62 | 6.85 | 121.50 | 0.67 | 0.67 | 7000k×256 | 250 |
| MDT-XL/2 (Gao et al., 2023) | 6.23 | 5.23 | 143.02 | 0.71 | 0.65 | 6500k×256 | 250 |
| **RDM** | **5.27** | 4.39 | 153.43 | 0.75 | 0.62 | 290k×4096 | 125 |
| ADM-U,G | 3.94 | 6.14 | 215.84 | 0.83 | 0.53 | 1980k×256 | 500 |
| LDM-4-G (CFG=1.50) | 3.60 | - | 247.67 | 0.87 | 0.48 | 178k×1200 | 500 |
| DiT-XL/2-G (CFG=1.50) | 2.27 | 4.60 | 278.24 | 0.83 | 0.57 | 7000k×256 | 500 |
| MDT-XL/2-G (dynamic CFG) | **1.79** | 4.57 | 283.01 | 0.81 | 0.61 | 6500k×256 | 500 |
| MDT-XL/2-G (CFG=1.325) | 2.26 | 4.28 | 246.06 | 0.81 | 0.59 | 6500k×256 | 500 |
| **RDM** (CFG=3.50) | 1.99 | 3.99 | 260.45 | 0.81 | 0.58 | 290k×4096 | 250 |
| + class-balance[1] | 1.87 | **3.97** | 278.75 | 0.81 | 0.59 | 290k×4096 | 250 |

StyleSwin (Zhang et al., 2022) with a remarkably fewer training iterations (50M versus 820M trained images). We also achieve the best precision and recall among the existing works.

**ImageNet** Table 3 shows the performance of class-conditional generative models on ImageNet $256 \times 256$. We report the best results as possible of the existing methods with classifier-free guidance (CFG) (Ho & Salimans, 2022). RDM achieves the best sFID and outperforms all the other methods by FID except MDT-XL/2 (Gao et al., 2023) with a dynamic CFG scale. If with a fixed but best-picked CFG scale[2], MDT-XL/2 can only achieve an FID of 2.26. While achieving competitive results, RDM is trained with only 70% of the iterations of MDT-XL/2 (1.2B versus 1.7B trained images), indicating that the longer training and a more granular CFG strategy are potential directions to further optimize the FID of RDM.

**Training Efficiency** We also compare the performance of RDM with existing methods along with the training cost in Figure 1. When CFG is disabled, RDM achieves a better FID than previous state-of-the-art diffusion models including DiT (Peebles & Xie, 2022) and MDT (Gao et al., 2023). RDM outperforms them even with only about $1/3$ training iterations.

## 4.3 ABLATION STUDY

In this section, we conduct ablation experiments on the designs of RDM to verify their effectiveness. Unless otherwise stated, we report results of RDM on $256 \times 256$ generation without CFG.

**The Effectiveness of block noise.** We compare the performance of RDM with and without adding block noise in Figure 4. With a sufficient phase of training, RDM with block noise outperforms the model without block noise by a remarkable margin on both ImageNet and CelebA-HQ. This demonstrates the effectiveness of the block noise. The addition of block noise introduces higher modeling complexity of the noise pattern, which contributes to a slower convergence of training in the initial stage, as illustrated by Figure 4(a). We assume that training on a significantly smaller scale of samples leads to a fast convergence of the model, which obliterates such a feature, therefore a similar phenomenon cannot be observed in the training of CelebA-HQ.

**The scale of stochasticity.** As previous works (Song et al., 2020b) have shown, SDE samplers usually perform better than ODE samplers. We want to quantitatively measure how the scale of the stochaticity affects the performance in the RDM sampler (Algorithm 1). Table 2 shows results with $\eta$ varying from 0 to 0.50. For both CelebA-HQ and ImageNet, the optimal FID is achieved by $\eta = 0.2$. We hypothesize a small $\eta$ is insufficient for the noise addition to cover the bias formed

---

[2]The best CFG scale is 1.325 with a hyperparameter sweep from 1.0 to 1.8. We observed the FID increases greatly if CFG scale > 1.5 for MDT-XL/2.

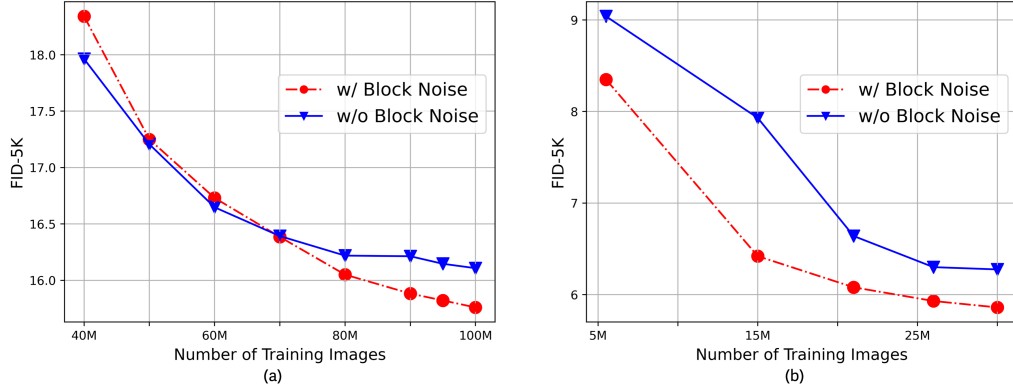

Figure 4: The effectiveness of block noise. We compare the performance of RDM along the training on (a) ImageNet $256 \times 256$ and (b) CelebA-HQ $256 \times 256$. To apply block noise in RDM, we set $\alpha = 0.15$ and kernel size $s = 4$.

in earlier sampling steps, while a large $\eta$ introduces excessive noise into the process of sampling, which makes a moderate $\eta$ to be the best choice. Within a reasonable scale of stochasticity, an SDE sampler always outperforms the ODE sampler by a significant margin.

**Sampling steps.** To show the efficiency of our model, we compare the performance of RDM and other methods with fewer sampling steps. Number of Function Evaluations (NFE), i.e., the number that a neural network is called during sampling, is used as the index of the comparison for fairness. For RDM, the NFE consists of the NFE in the second stage and $1/10$ the NFE in the first stage, according to the proportion of the FLOPs. As shown in Figure 5, the performance of DiT-XL/2 and MDT-XL/2 both drop significantly with a lower NFE, while RDM barely declines. Considering that the steps in different stages may contribute differently in FID, we demonstrates three FLOPs allocation strategies in Figure 5. With more NFE allocated in the first stage, RDM achieves a better FID. In all settings, RDM performs better than MDT-XL/2 and DiT-XL/2 if NFE < 200.

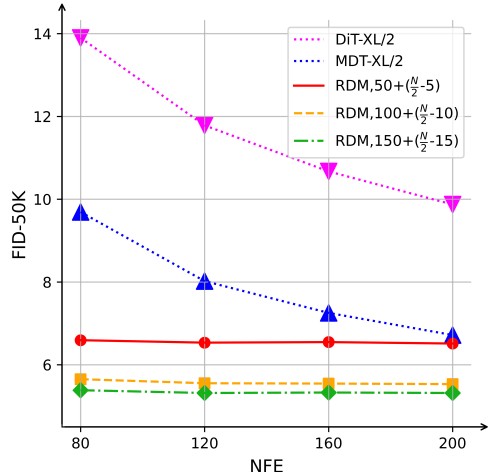

Figure 5: Comparison of FID on ImageNet with different sampling steps. For allocation of NFE $= N$ in RDM, $10n + \left(\frac{N}{2} - n\right)$ means $10n$ for the first stage and $\frac{N}{2} - n$ for the second.

## 5 CONCLUSION AND DISCUSSION

In this paper, we propose relay diffusion to optimize the cascaded pipeline. The diffusion process can now continue when changing the image resolution or model architectures. We anticipate that our method can reduce the cost of training and inference, and help create more advanced text-to-image model in the future.

The frequency analysis in the paper reveals the relation between noise and image resolution, which might be helpful to design a better noise schedule. However, our numerous attempts to theoretically derive the optimal noise schedule on the dataset from a frequency perspective did not yield good results. The reason might be that the optimal noise schedule is also related to the size of the model, inductive bias, and the nuanced distribution characteristics of the data. Further investigation is left for future work.

ACKNOWLEDGMENTS

This work is supported by Technology and Innovation Major Project of the Ministry of Science and Technology of China under Grant 2022ZD0118600, the NSFC for Distinguished Young Scholar 61825602, Tsinghua University Initiative Scientific Research Program 20233080067 and the New Cornerstone Science Foundation through the XPLORER PRIZE. The authors also thank Ting Chen from Google DeepMind and Junbo Zhao from Zhejiang University for their valuable talks and comments.

AUTHOR CONTRIBUTIONS

Ming Ding proposes the methods and leads the project. Jiayan Teng and Wendi Zheng conduct most of the experiments. Wenyi Hong works together on early experiments. Jianqiao Wangni, Wenyi Hong and Zhuoyi Yang contribute to the writing of the paper. Jie Tang provides guidance and supervision.

The work is partly done during the internship of Jiayan Teng and Wendi Zheng at Zhipu AI.

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

# A DERIVATION

## A.1 PATCH-WISE BLURRING

The forward process of blurring diffusion is defined as Eq. 3, where $\boldsymbol{u}_0 = \boldsymbol{V}^{\mathrm{T}}\boldsymbol{x}_0$ denotes the representation of the image $\boldsymbol{x}_0$ in the frequency space. The diagonal matrix $\boldsymbol{D}_t = e^{\boldsymbol{\Lambda}t}$ defines a non-isotropic blurring projection, where $\boldsymbol{\Lambda}(i \times W + j, i \times W + j) = -\pi^2(\frac{i^2}{H^2} + \frac{j^2}{W^2})$ corresponds to the coordinate $(i, j)$ in the 2D frequency space. In the equation $q(\boldsymbol{u}_t|\boldsymbol{u}_0) = \mathcal{N}(\boldsymbol{u}_t|\boldsymbol{D}_t\boldsymbol{u}_0, \sigma_t^2\boldsymbol{I})$, we can utilize the dot product of matrices to transform $\boldsymbol{D}_t$ and $\boldsymbol{u}_0$ into 2D matrices, $\tilde{\boldsymbol{D}}_t$ and $\tilde{\boldsymbol{u}}_0$, in the shape of $H \times W$ for calculation:

$$\boldsymbol{D}_t\boldsymbol{u}_0 \Rightarrow \tilde{\boldsymbol{D}}_t \cdot \tilde{\boldsymbol{u}}_0 \tag{12}$$

In the super-resolution stage of RDM, we apply blurring on each $k \times k$ patch independently. We name it as patch-wise blurring and define the diagonal blurring matrix in the shape of $k \times k$ for each patch as:

$$\tilde{\boldsymbol{D}}_{t,k \times k} = \exp(\tilde{\boldsymbol{\Lambda}}_{k \times k}t), \quad \tilde{\boldsymbol{\Lambda}}_{k \times k}(i, j) = -\pi^2(\frac{i^2}{k^2} + \frac{j^2}{k^2}), \tag{13}$$

where $i \in [0, k), j \in [0, k)$. For any patch, $\tilde{\boldsymbol{D}}_{t,k \times k}$ remains the same. The blurring matrix $\tilde{\boldsymbol{D}}_t^p$ of the patch-wise blurring is a combination of all the independent blurring matrices $\tilde{\boldsymbol{D}}_{t,k \times k}$. The relationship between the elements of $\tilde{\boldsymbol{D}}_t^p$ and $\tilde{\boldsymbol{D}}_{t,k \times k}$ can be expressed as:

$$\tilde{\boldsymbol{D}}_t^p(i, j) = \tilde{\boldsymbol{D}}_{t,k \times k}(i \bmod k, j \bmod k), \tag{14}$$

where $(i, j)$ corresponds to the coordinate in the 2D frequency space. Finally, $\boldsymbol{D}_t^p$ in Eq. 7 can be formulated as:

$$\boldsymbol{D}_t^p = \mathrm{diag}(\mathrm{unfold}(\tilde{\boldsymbol{D}}_t^p)), \tag{15}$$

where $\mathrm{unfold}(\tilde{\boldsymbol{D}}_t^p)$ means unfolding the $H \times W$ matrix into a vector of $HW$ dimensions and $\mathrm{diag}(\boldsymbol{v})$ denotes the diagonal matrix with vector $\boldsymbol{v}$ as its diagonal line.

## A.2 COMBINATION OF SCHEDULE

We follow Karras et al. (2022) to set the noise schedule for standard diffusion as $\ln(\sigma) \sim \mathcal{N}(P_{mean}, P_{std}^2)$. We use $\mathcal{F}_{\mathcal{D}}$ and $\mathcal{F}_{\mathcal{D}}^{-1}$ to denote the cumulative distribution function (CDF) and the inverse distribution function (IDF) for distribution $\mathcal{D}$ in the following description. With $t$ sampled from uniform distribution $\mathcal{U}(0, 1)$, the noise scale is formulated as:

$$\sigma(t) = \exp(\mathcal{F}_{\mathcal{N}(P_{mean}, P_{std}^2)}^{-1}(t)). \tag{16}$$

For the super-resolution stage of RDM, we apply a truncated version of diffusion noise schedule $\sigma'(t), t \sim \mathcal{U}(0, 1)$. If we set $t_s$ as the starting point of the truncation, the new noise schedule can be formally expressed as:

$$\sigma'(t) = \sigma(\mathcal{F}_{\mathcal{U}(0,1)}^{-1}(\mathcal{F}_{\mathcal{U}(0,1)}(t_s)\mathcal{F}_{\mathcal{U}(0,1)}(t))), \tag{17}$$

which means we only sample the noise scale $\sigma'$ from positions of the normal distribution $\mathcal{N}(P_{mean}, P_{std}^2)$ where its CDF is less than $t_s$.

For the process of blurring, we set its schedule following the setting of Hoogeboom & Salimans (2022). They found that the heat dissipation is equivalent to a Gaussian blur with the variance of its kernel as $\sigma_{B,t}^2 = 2\tau_t$. They set the blurring scale $\sigma_{B,t}$ as:

$$\sigma_{B,t} = \sigma_{B,max}\sin^2(\frac{t\pi}{2}), \tag{18}$$

where $t$ is also sampled from the uniform distribution $\mathcal{U}(0, 1)$ and $\sigma_{B,max}$ denotes a fixed hyperparameter. Empirically, we set $\sigma_{B,max} = 3$ for ImageNet $256 \times 256$ and $\sigma_{B,max} = 2$ for CelebA-HQ $256 \times 256$. The blurring matrix is formulated as $\boldsymbol{D}_t = e^{\boldsymbol{\Lambda}\tau_t}$, where $\tau_t = \frac{\sigma_{B,t}^2}{2}$. As illustrated in Section 2.2, $\boldsymbol{\Lambda}$ is a diagonal matrix and $\boldsymbol{\Lambda}_{i \times W + j} = -\pi^2(\frac{i^2}{H^2} + \frac{j^2}{W^2})$ for coordinate $(i, j)$.

### A.3 SAMPLER DERIVATION

In this section, we prove the consistency between the design of our sampler and the formulation of blurring diffusion. We need to prove that the jointly distribution $q_\delta(\boldsymbol{u}_{n-1}|\boldsymbol{u}_n,\boldsymbol{u}_0)$ we define in Eq. 10 matches with the marginal distribution

$$q_\delta(\boldsymbol{u}_n|\boldsymbol{u}_0) = \mathcal{N}(\boldsymbol{u}_n|\boldsymbol{D}^p_{t_n}\boldsymbol{u}_0, \sigma^2_{t_n}\boldsymbol{I}) \tag{19}$$

under the condition of Eq. 9.

*proof.* Given that $q_\delta(\boldsymbol{u}_N|\boldsymbol{u}_0) = \mathcal{N}(\boldsymbol{u}_N|\boldsymbol{D}^p_{t_N}\boldsymbol{u}_0, \sigma^2_{t_N}\boldsymbol{I})$, we proceed with a mathematical induction approach. Assuming that for any $n \le N$, $q_\delta(\boldsymbol{u}_n|\boldsymbol{u}_0) = \mathcal{N}(\boldsymbol{u}_n|\boldsymbol{D}^p_{t_n}\boldsymbol{u}_0, \sigma^2_{t_n}\boldsymbol{I})$ holds. We only need to prove $q_\delta(\boldsymbol{u}_{n-1}|\boldsymbol{u}_0) = \mathcal{N}(\boldsymbol{u}_{n-1}|\boldsymbol{D}^p_{t_{n-1}}\boldsymbol{u}_0, \sigma^2_{t_{n-1}}\boldsymbol{I})$, and then the conclusion above will be proved based on the induction hypothesis.

Firstly, based on

$$q_\delta(\boldsymbol{u}_{n-1}|\boldsymbol{u}_0) = \int q_\delta(\boldsymbol{u}_{n-1}|\boldsymbol{u}_n,\boldsymbol{u}_0)q(\boldsymbol{u}_n|\boldsymbol{u}_0)d\boldsymbol{u}_n, \tag{20}$$

we introduce

$$q_\delta(\boldsymbol{u}_{n-1}|\boldsymbol{u}_n,\boldsymbol{u}_0) = \mathcal{N}\left(\boldsymbol{u}_{n-1}|\frac{1}{\sigma_{t_n}}(\sqrt{\sigma^2_{t_{n-1}}-\delta^2_n}\boldsymbol{u}_n + (\sigma_{t_n}\boldsymbol{D}^p_{t_{n-1}} - \sqrt{\sigma^2_{t_{n-1}}-\delta^2_n}\boldsymbol{D}^p_{t_n})\boldsymbol{u}_0), \delta^2_n\boldsymbol{I}\right) \tag{21}$$

and

$$q_\delta(\boldsymbol{u}_n|\boldsymbol{u}_0) = \mathcal{N}(\boldsymbol{u}_n|\boldsymbol{D}^p_{t_n}\boldsymbol{u}_0, \sigma^2_{t_n}\boldsymbol{I}). \tag{22}$$

Then according to Bishop & Nasrabadi (2006), $q_\delta(\boldsymbol{u}_{n-1}|\boldsymbol{u}_0)$ is also a Gaussian distribution:

$$q_\delta(\boldsymbol{u}_n|\boldsymbol{u}_0) = \mathcal{N}(\boldsymbol{u}_n|\boldsymbol{\mu}_{n-1}, \boldsymbol{\Sigma}_{n-1}). \tag{23}$$

Therefore, from Eq. 20, we can derive that

$$\boldsymbol{\mu}_{n-1} = \frac{1}{\sigma_{t_n}}(\sqrt{\sigma^2_{t_{n-1}}-\delta^2_n}\boldsymbol{D}^p_{t_n}\boldsymbol{u}_0 + (\sigma_{t_n}\boldsymbol{D}^p_{t_{n-1}} - \sqrt{\sigma^2_{t_{n-1}}-\delta^2_n}\boldsymbol{D}^p_{t_n})\boldsymbol{u}_0) = \boldsymbol{D}^p_{t_{n-1}}\boldsymbol{u}_0 \tag{24}$$

and

$$\boldsymbol{\Sigma}_{n-1} = \frac{\sigma^2_{t_{n-1}}-\delta^2_n}{\sigma^2_{t_n}}\sigma^2_{t_n}\boldsymbol{I} + \delta^2_n\boldsymbol{I} = \sigma^2_{t_{n-1}}\boldsymbol{I}. \tag{25}$$

Summing up, $q_\delta(\boldsymbol{u}_{n-1}|\boldsymbol{u}_0) = \mathcal{N}(\boldsymbol{u}_{n-1}|\boldsymbol{D}^p_{t_{n-1}}\boldsymbol{u}_0, \sigma^2_{t_{n-1}}\boldsymbol{I})$. The inductive proof is complete.

## A.4 STOCHASTIC SAMPLER

---

**Algorithm 1** the RDM second-order stochastic sampler

---

$\quad$ **sample** $\hat{\boldsymbol{x}}_N$ from results of the first stage $\qquad\qquad$ ▷ i.e. images at the resolution of $64 \times 64$

$\quad \boldsymbol{x}_N = interpolate(\hat{\boldsymbol{x}}_N, 256, mode = "nearest") + \sigma_{t_n}\tilde{\boldsymbol{\epsilon}}$ $\quad$ ▷ upsample 64px images to blurry 256px images and apply the truncated schedule in Appendix A.2

$\quad \boldsymbol{u}_N = \boldsymbol{V}^{\mathrm{T}}\boldsymbol{x}_N$ $\qquad\qquad\qquad\qquad\qquad\qquad\qquad\qquad$ ▷ transformed into the frequency domain

$\quad$ **for** $n \in \{N, \ldots, 1\}$ **do**

$\qquad \gamma_n = \sqrt{1 - \eta^2}\frac{\sigma_{t_{n-1}}}{\sigma_{t_n}}, \quad \delta_n = \eta\sigma_{t_{n-1}}$ $\qquad\qquad\qquad\qquad$ ▷ coefficient of the random term

$\qquad \tilde{\boldsymbol{u}}_0 = \boldsymbol{u}_\theta(\boldsymbol{u}_n, \sigma_{t_n})$ $\qquad\qquad\qquad\qquad\qquad\qquad\qquad$ ▷ model prediction at $t_n$

$\qquad \boldsymbol{d}_n = \frac{\boldsymbol{u}_n - \tilde{\boldsymbol{u}}_0}{\sigma_{t_n}}$ $\qquad\qquad\qquad\qquad\qquad\qquad\qquad$ ▷ first-order gradient term at $t_n$

$\qquad \boldsymbol{u}_{n-1} = (\boldsymbol{D}_{t_{n-1}}^p + \gamma_n(\boldsymbol{I} - \boldsymbol{D}_{t_n}^p))\boldsymbol{u}_n + \sigma_{t_n}(\gamma_n\boldsymbol{D}_{t_n}^p - \boldsymbol{D}_{t_{n-1}}^p)\boldsymbol{d}_n + \delta_n\tilde{\boldsymbol{\epsilon}}$

$\qquad\qquad\qquad\qquad\qquad\qquad\qquad\qquad\qquad\qquad$ ▷ from $t_n$ to $t_{n-1}$ using Euler's method

$\qquad$ **if** $n \neq 1$ **then** $\qquad\qquad\qquad\qquad\qquad\qquad\qquad\qquad$ ▷ the second-order part

$\qquad\qquad \tilde{\boldsymbol{u}}_0' = \boldsymbol{u}_\theta(\boldsymbol{u}_{n-1}, \sigma_{t_{n-1}})$ $\qquad\qquad\qquad\qquad$ ▷ model prediction at $t_{n-1}$

$\qquad\qquad \boldsymbol{d}_{n-1} = \frac{\boldsymbol{u}_{n-1} - \tilde{\boldsymbol{u}}_0'}{\sigma_{t_{n-1}}}$ $\qquad\qquad\qquad\qquad\qquad$ ▷ gradient term at $t_{n-1}$

$\qquad\qquad \boldsymbol{d}_n' = \frac{\boldsymbol{d}_n + \boldsymbol{d}_{n-1}}{2}$ $\qquad\qquad\qquad\qquad\qquad$ ▷ second-order gradient term

$\qquad\qquad \boldsymbol{u}_{n-1}' = (\boldsymbol{D}_{t_{n-1}}^p + \gamma_n(\boldsymbol{I} - \boldsymbol{D}_{t_n}^p))\boldsymbol{u}_n + \sigma_{t_n}(\gamma_n\boldsymbol{D}_{t_n}^p - \boldsymbol{D}_{t_{n-1}}^p)\boldsymbol{d}_n' + \delta_n\tilde{\boldsymbol{\epsilon}}$ $\qquad$ ▷ correction

$\qquad$ **end if**

$\qquad \boldsymbol{u}_{n-1} = \boldsymbol{u}_{n-1}'$

$\quad$ **end for**

$\quad \boldsymbol{x}_0 = \boldsymbol{V}\boldsymbol{u}_0$

---

As for the sampler of the first stage, we follow the EDM sampler (Karras et al., 2022). Of course, samplers such as DDPM are also capable. After all, the first stage is just a standard diffusion model.

## B MODEL DETAILS

### B.1 HYPERPARAMETERS

Hyperparameters we use for the training of RDM are presented in Table 4. We set the architecture hyperparameters for diffusion models following Dhariwal & Nichol (2021), in corresponding to the input resolutions. For the experiments on CelebA-HQ, we set the model dropout to be larger (0.15 and 0.2 for two stages respectively), and enable sample augmentation to prevent RDM from overfitting.

### B.2 TRAINING COST

On ImageNet, the first stage model was trained on 32 V100 for 13 days according to EDM (Karras et al., 2022) and the second stage model ($64 \to 256$) was trained on 64 40G-A100 for 12.5 days. On CelebA-HQ, we trained the first stage model on 32 40G-A100 for 16 hours and the second stage model ($64 \to 256$) on 32 40G-A100 for 25.5 hours.

## C ADDITIONAL EXPERIMENTS

### C.1 FURTHER EXPERIMENTS ON HIGHER RESOLUTIONS

To further show that RDM can easily scale to high-resolution image generation without carefully adjusting the hyperparameters of noise schedule and architecture. We take dataset CelebA-HQ as an example and conduct experiments on higher resolutions: 512 and 1024. Except for the different hyperparameters shown in Table 5, other settings remain the same as the 64→256 model in the second stage.

Table 4: Hyperparameters for RDM.

|  | ImageNet 64 | ImageNet 64→256 | CelebA-HQ 64 | CelebA-HQ 64→256 |
|---|---|---|---|---|
| Diffusion steps | 256 | 100 | 120 | 53 |
| Model size | 295M | 553M | 295M | 553M |
| GFLOPs | 104 | 1117 | 104 | 1117 |
| Mixed-precision (FP16) | ✓ | ✓ | - | ✓ |
| Channels | 192 | 256 | 192 | 256 |
| Channels multiple | 1,2,3,4 | 1,1,2,2,4,4 | 1,2,3,4 | 1,1,2,2,4,4 |
| Heads Channels | 64 | 64 | 64 | 64 |
| Attention resolution | 32,16,8 | 32,16,8 | 32,16,8 | 32,16,8 |
| Dropout | 0.1 | 0.1 | 0.15 | 0.2 |
| Augment probability | 0 | 0 | 0.2 | 0.2 |
| Blurring $\sigma_{max}$ | - | 3.0 | - | 2.0 |
| Batch size | 4096 | 4096 | 1024 | 1024 |
| Training Images | 2500M | 1000M | 70M | 40M |
| Learning Rate | 1e-4 | 1e-4 | 1e-4 | 1e-4 |

Table 5: Hyperparameters for RDM on higher resolutions of 512 and 1024.

|  | CelebA-HQ 256→512 | CelebA-HQ 256→1024 |
|---|---|---|
| Diffusion steps | 35 | 35 |
| Model size | 558M | 562M |
| GFLOPs | 1987 | 4509 |
| Mixed-precision (FP16) | ✓ | ✓ |
| Channels | 256 | 256 |
| Channels multiple | 0.5,1,1,2,2,4,4 | 0.5,0.5,1,1,2,2,4,4 |
| Heads Channels | 64 | 64 |
| Attention resolution | 32,16,8 | 32,16,8 |
| Dropout | 0.2 | 0.1 |
| Augment probability | 0.2 | 0.2 |
| Blurring $\sigma_{max}$ | 1.25 | 2.0 |
| Batch size | 256 | 256 |
| Training Images | 15M | 11M |
| Learning Rate | 1e-4 | 1e-4 |

The cost in Table 6 and Table 7 contains the training of three stages, with the same equivalent conversion as Table 1 and Table 3, according to GFLOPs. As shown in the table, RDM achieves state-of-the-art FID at the resolution of both 512 and 1024, and only requires a small amount of training in the third stage according to Table 5. This demonstrates that RDM can be easily extended from two stages to three stages and higher resolutions. Examples are shown in Figure 12.

Table 6: Benchmarking unconditional image generation on CelebA-HQ $512 \times 512$.

| **Unconditional CelebA-HQ** $512 \times 512$ | | |
|---|---|---|
| Model | FID↓ | Cost(Number of Training Images) |
| WaveDiff (Phung et al., 2023) | 6.40 | 12M |
| **RDM** | **3.47** | 41M |

## C.2 ADD BLOCK NOISE TO END-TO-END DIFFUSION MODEL

To further demonstrate the effectiveness of block noise, we conduct ablation experiments on an end-to-end model followed by the setting of ADM (Dhariwal & Nichol, 2021). We use a mixture of

Table 7: Benchmarking unconditional image generation on CelebA-HQ $1024 \times 1024$.

| Unconditional CelebA-HQ $1024 \times 1024$ | | |
|---|---|---|
| Model | FID↓ | Cost(Number of Training Images) |
| HiT-B (Zhao et al., 2021) | 8.83 | 16M |
| Polarity-ProGAN (Humayun et al., 2022) | 7.28 | - |
| StyleGAN (Karras et al., 2019) | 5.06 | - |
| StyleSwin (Zhang et al., 2022) | 4.43 | 819M |
| **RDM** | **3.85** | 23M |

fixed ratio block noise and Gaussian noise as illustrated in section 3.2. As shown in Figure 6, ADM with block noise outperforms the model without block noise by a remarkable margin on CelebA-HQ $256 \times 256$.

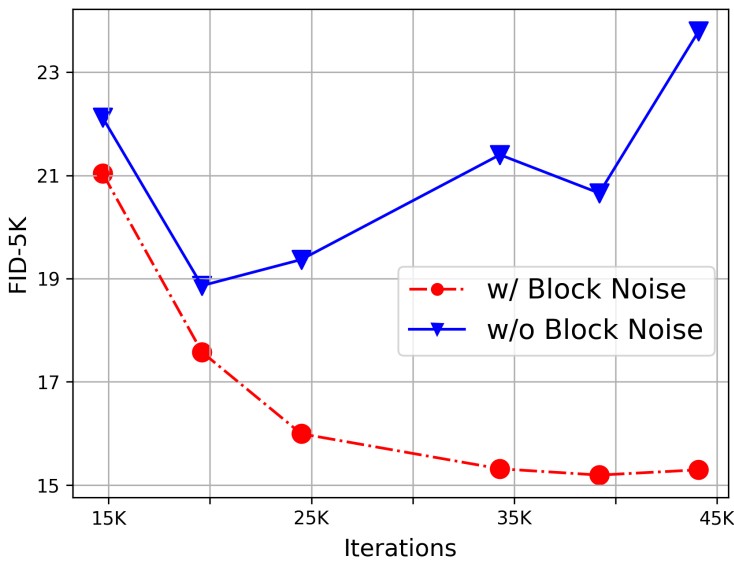

Figure 6: The effectiveness of block noise. We compare the performance of ADM along the training on CelebA-HQ $256 \times 256$. For block noise, we set $\alpha = 0.15$ and kernel size $s = 4$.

### C.3 Correct Artifacts in the Low-resolution Images

As illustrated in section 3.2, the super-resolution generation of RDM is a Markovian process, in comparison of CDM and ADM-U using low-resolution conditioning all along the generation process. This could improve the robustness of RDM on handling artifacts from low-resolution stages. Figure 7 shows the comparison of super-resolution generation between RDM and ADM-U. We use $64 \times 64$ samples from ImageNet as low-resolution inputs, adding 0.05 scale of Gaussian Noise to introduce artifacts. While RDM successfully handles the noise to generate clean $256 \times 256$ samples, ADM-U preserves the noise to the $256 \times 256$ samples.

## D  Details About The Power Spectral Density

### D.1 Calculation Procedure of the PSD

We follow the setting of Rissanen et al. (2022) to calculate the PSD in the frequency space. The PSD at a certain frequency is defined as the square of the DCT coefficient at that frequency. Firstly,

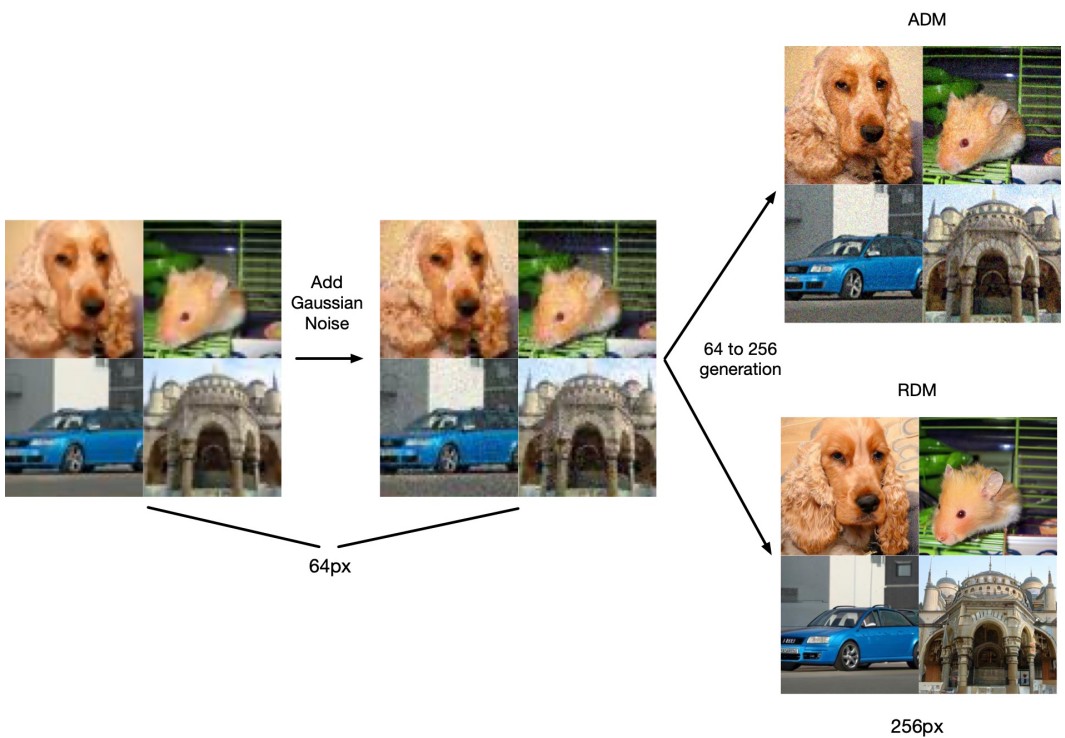

Figure 7: Input 64px images with artifacts into the 64→256 model of RDM (right top) and ADM-U (right bottom) to generate 256px images.

we transform the image into the 2D frequency space by DCT and set the frequency range to $[0, \pi]$. To obtain the 1D curve of the PSD, we calculate the distance from each point $(x, y)$ to the origin in the frequency space, i.e. $\sqrt{x^2 + y^2}$, considering it as a 1D frequency value. Subsequently, we uniformly divide the frequency domain into $N$ intervals, and take the midpoint of each interval as its representative frequency value. Finally, we take the mean of the PSD values for all points within the interval as the PSD value for that interval, in order to get $N$ coordinate pairs for plotting. The SNR curve in Figure 2 can be obtained in a similar approach, while the only difference is that the vertical axis values are replaced with the absolute value of the ratio between the DCT coefficients for the image and noise in the frequency space.

## D.2 Analysis of the PSD

As shown in Figure 8, the PSD of real images gradually decreases from low frequency to high frequency. And the intensity of Gaussian noise components across all frequency bands is generally equal. Therefore, when corrupting real images, Gaussian noise initially drowns out high-frequency components until the noise intensity becomes high enough to drown out the low-frequency components of real images. And it is demonstrated in Figure 2 that, as the resolution of images increases, less information is corrupted under the same noise intensity. Correspondingly, as shown in Figure 8(a) and Figure 8(b), the low-frequency portion of the PSD gets drowned out more slowly as the resolution increases. It is indicated that we will introduce excessive high-frequency components of noise when corrupting the low-frequency information of real images, especially for high-resolution images.

Differently, the low-frequency portion of the PSD from block noise is notably higher than that of Gaussian noise with the same intensity. Furthermore, the PSD of block noise exhibits a decreasing trend as frequency increases, and its curve is quite similar to the PSD curve of Gaussian noise at the resolution of 64 upsampled to the resolution of 256. This leads to the PSD curves of high-resolution images with added block noise and that of low-resolution images with added Gaussian noise also

being quite similar. As a result, the low-frequency portion of the PSD from images with added block noise gets drowned out more quickly than that from images with added Gaussian noise. We can conclude that block noise can corrupt the low-frequency components of images more easily.

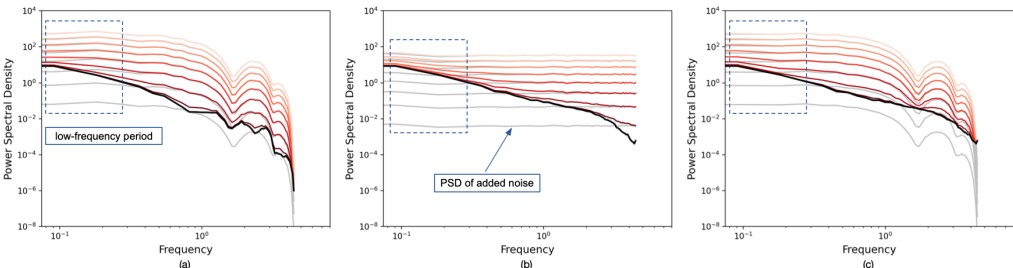

Figure 8: The power spectral density (PSD) of real images after adding (a) 64px Gaussian noise, (b) 256px Gaussian noise and (c) 256px block noise with block size of 4. The black curve represents the PSD of real images. The red curves, from dark to light, represent adding noise with increasing intensity. In order to make comparisons within the same frequency space, for the images at the resolution of 64, we firstly upsample them to the pixel space at the resolution of 256.

## E    ADDITIONAL SAMPLES

Section 4.3 quantitatively compares the performance of RDM with other models under the same NFE and demonstrates the superiority of RDM with fewer sampling steps. Figure 9 shows qualitative comparison results. While other models achieve competitive quality of generation with sufficient NFE, their performances degenerate noticeably with the decrease of NFE. In contrast, RDM still maintains comparable generation quality with a low NFE.

Figure 10 compares visualized samples generated by the best settings of StyleGAN-XL (Sauer et al., 2022), DiT (Peebles & Xie, 2022) and RDM. StyleGAN-XL is in the framework of GAN, while DiT and RDM are diffusion models. RDM achieves the best quality of images synthesis. Figure 11 exhibits more examples generated by our model RDM on ImageNet $256 \times 256$.

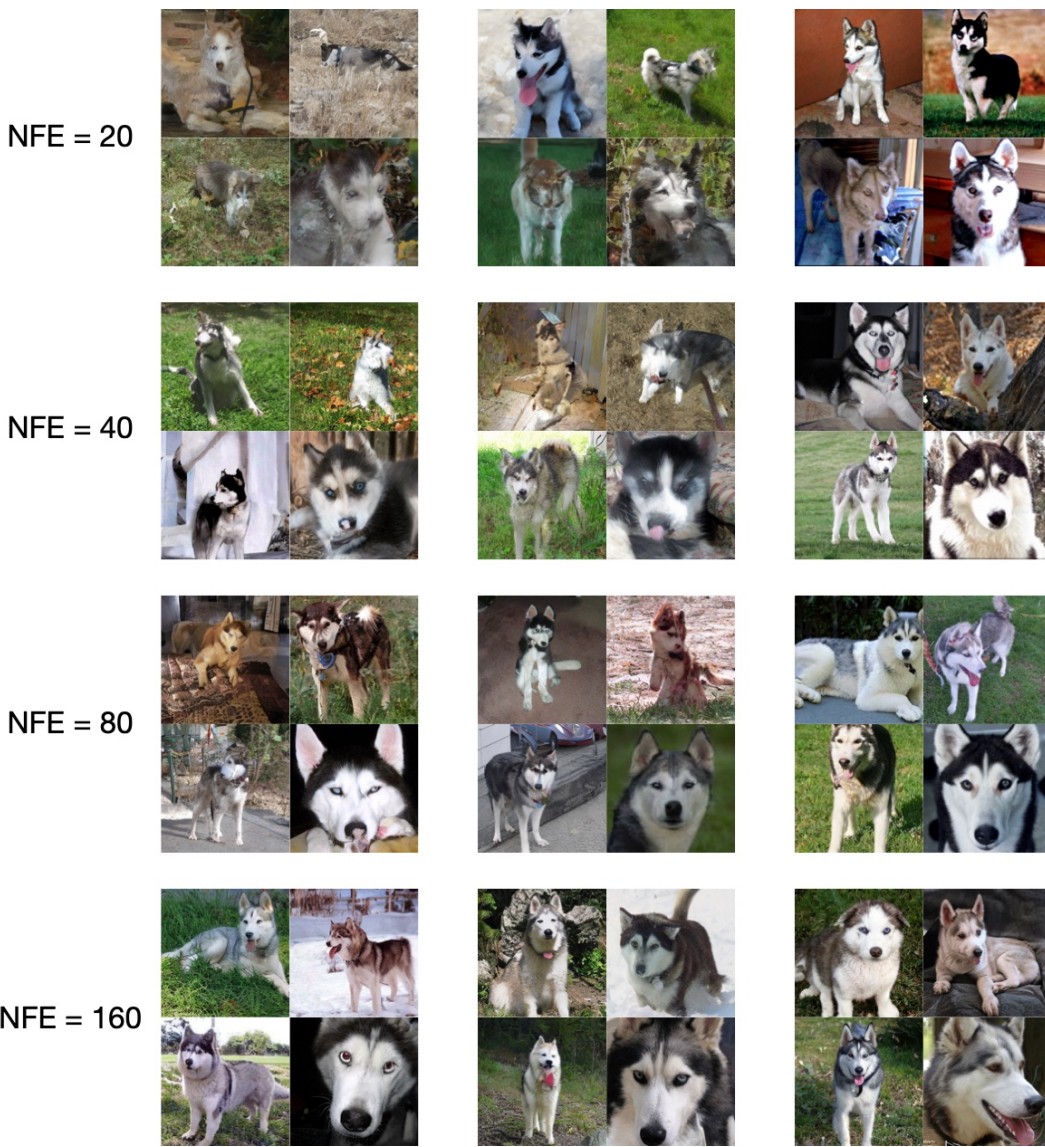

Figure 9: Comparison of ImageNet samples with varied NFE. DiT-XL/2 (left) vs MDT-XL/2 (middle) vs RDM (right). The allocation of NFE between the two stages of RDM is: [2, 18], [8, 32], [20, 60], [40, 120].

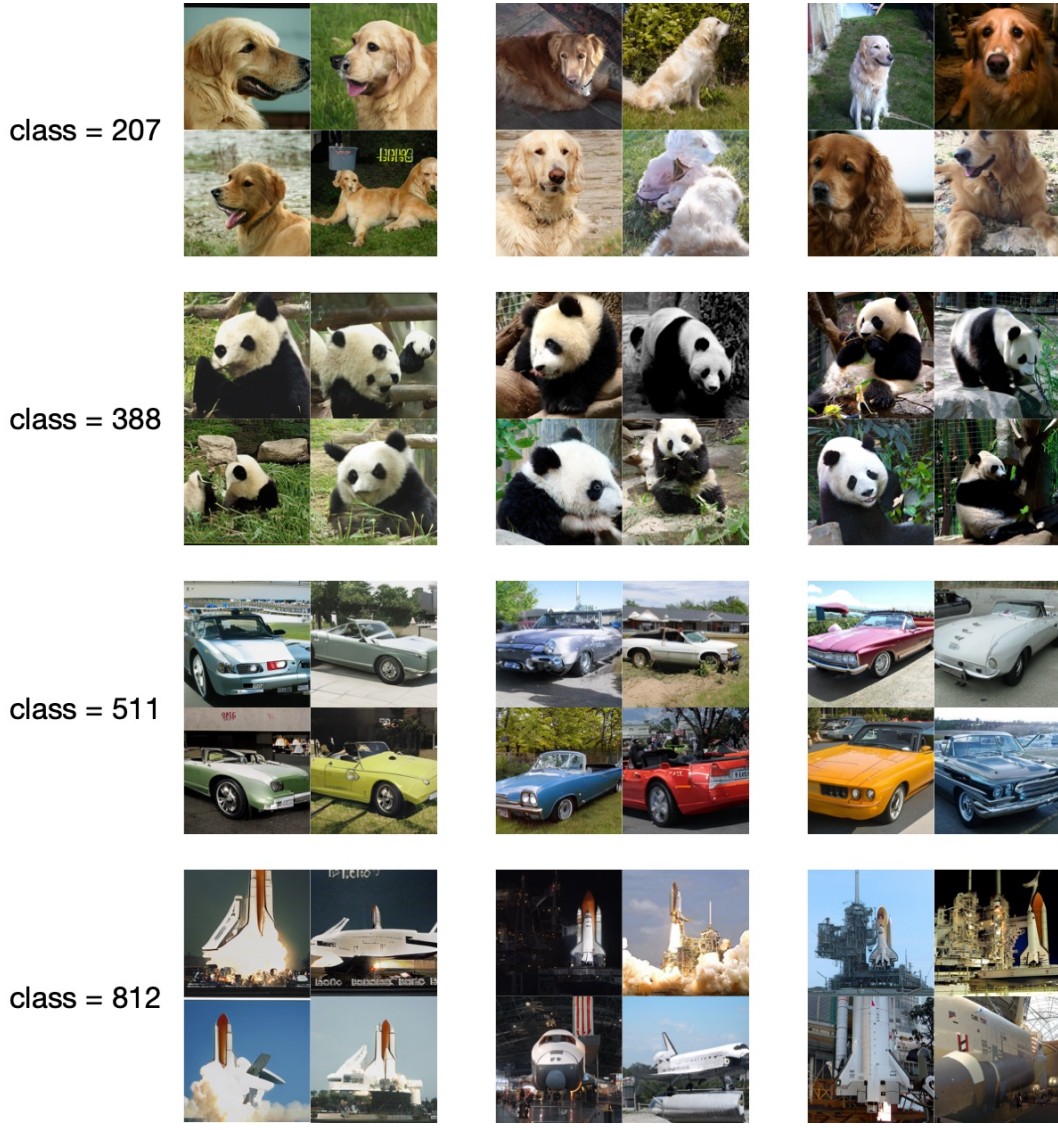

Figure 10: Comparison of best ImageNet samples. StyleGAN-XL (FID 2.30, left) vs DiT-XL/2 (FID 2.27, middle) vs RDM (FID 1.87, right).

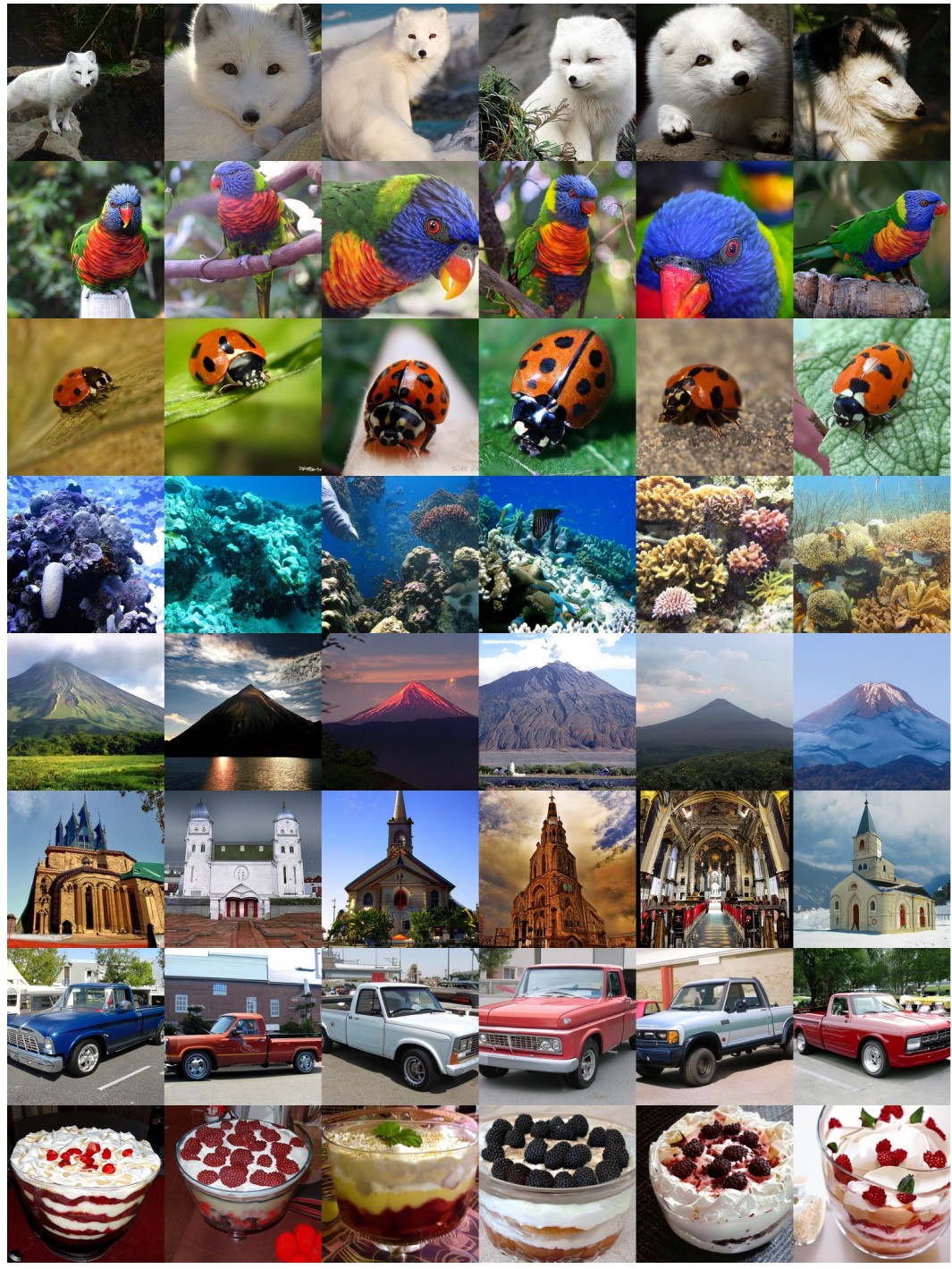

Figure 11: Additional ImageNet samples generated by RDM. Classes are 279: Arctic fox, 90: lorikeet, 301: ladybug, 973: coral reef, 980: volcano, 497: church, 717: pickup truck, 927: trifle.

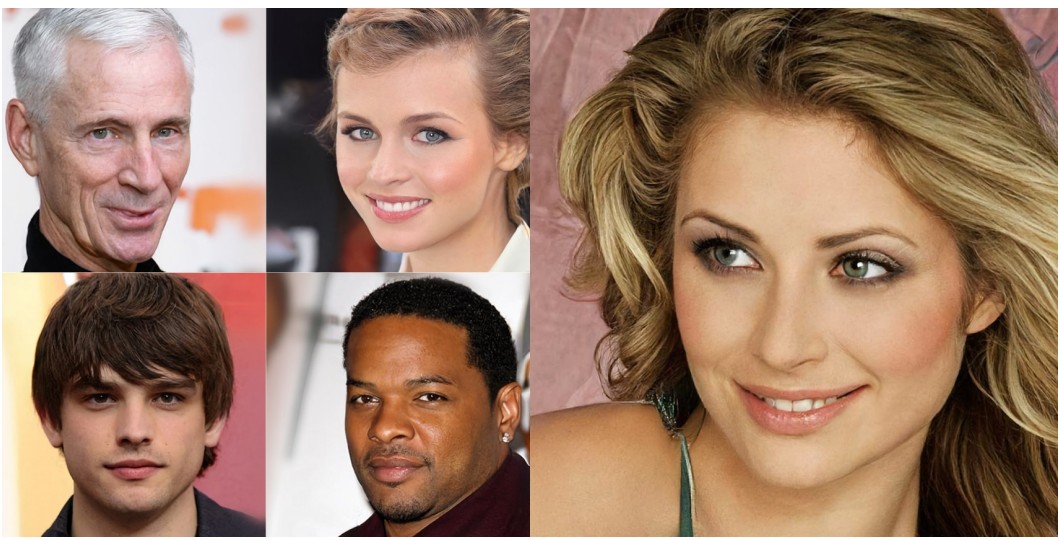

Figure 12: Samples generated by RDM on CelebA-HQ $512 \times 512$ (left) and CelebA-HQ $1024 \times 1024$ (right).

