# OpenReview forum: "Relay Diffusion: Unifying diffusion process across resolutions for image synthesis"
_ICLR.cc/2024/Conference — ICLR 2024 spotlight_

### Official Review · Reviewer_edjv · 2023-10-16

**Soundness:** 3 good
**Presentation:** 3 good
**Contribution:** 3 good
**Rating:** 6
**Confidence:** 4

**Summary:**

This paper proposes a Relay Diffusion Model (RDM) with the aid of block noise to overcome the drawback of the existing cascaded models. The proposed method is claimed to reduce the training and sampling steps. The experiments show that RDM outperforms other methods on FID and sFID on two datasets CelebA-HQ and ImageNet with resolutions of 256x256.

**Strengths:**

+ The finding that the same noise level can give a mismatch between low- and high-resolution images is interesting.
+ The design of the method can facilitate the training speed where it avoids the conditioning on low-resolution images.
+ Some improvements are sound

**Weaknesses:**

+ The paper mentioned that "most current models follow the linear (Ho et al., 2020) or cosine (Nichol & Dhariwal, 2021) schedule" and "an ideal noise schedule should be resolution-dependent (See Figure 2)" but looking into Figure 2, I cannot see which noise schedule is used there, either cosine or linear, making it very confusing to capture what the authors want to say. Again, the next sentence said "train high-resolution models directly with common schedules designed for resolutions of 32×32 or 64×64 pixels" but Figure 2 only shows the 64px and 256px (what about 32x32) --> What is the relationship between 32x32 or 64x64 pixels and the one shown in Figure 2 (64px/256px)?

+ The advantage of the proposed method is emphasized with training efficiency, however, it only considers the setting without CFG where some competitors do not yield the optimal output. I believe that it is better also more meaningful to compare all methods in the setting with CFG where the existing methods achieve the best performance (optimal setting). Furthermore, when claiming the efficiency, I would also expect the comparison of the inference time/steps of the proposed method with the existing ones in their best optimal regime (including sampling steps in Table 3.).

+ While the proposed method achieves slightly better sFID compared to MDT-XL/2-G on class-conditional generation ImageNet 256x256 (Table 3) (3.97 vs. 4.57, not really "a large margin" as stated in the abstract), however, most other metrics lag behind MDT-XL/2-G (such as FID, IS, and Recall). This indicates that to show its advantages over the existing approaches, it may need to present more evidence.

+ Section 4.2 talking about CeleA-HQ, mentioned fewer training iterations while it is given with 50M and 820M trained images, making it a bit confusing that the dataset contained a hundred million images or that is just the total training steps. I recommend revising it more clearly and putting a column on the side of that table about the total training iterations or total training images.

``[Post rebuttal update]: After the author's response, somehow it resolved some of my concerns, I increased my rating score from 5 to 6.``

**Questions:**

1) What is the specific of each group in Table 3? It shows that the first column (Model) presents 5 groups (separated by \midrule) but unclear why is it divided like that.

2) I wondering if the number of data used in this paper is wrong, for example, Table 4 stated that 2500M training images were used for ImageNet64, is that 2.5 billion images? To my knowledge, ImageNet contains more than one million images, am I wrong? The same for all other datasets., does CelebA-HQ have 70 million images for training? Also, in Figure 1, is the horizontal axis the real number of images for training or it is just the iterations?

---

> ### Author Response · Authors · 2023-11-17
> **Response to Reviewer edjv (1/N)**
>
> Thank you for your valuable comments, we will explain your concerns point by point.
>
> ======================== For Weakness ========================
>
> > **Weakness 1:** The paper mentioned that "most current models follow the linear (Ho et al., 2020) or cosine (Nichol & Dhariwal, 2021) schedule" and "an ideal noise schedule should be resolution-dependent (See Figure 2)" but looking into Figure 2, I cannot see which noise schedule is used there, either cosine or linear, making it very confusing to capture what the authors want to say. Again, the next sentence said "train high-resolution models directly with common schedules designed for resolutions of 32×32 or 64×64 pixels" but Figure 2 only shows the 64px and 256px (what about 32x32) --> What is the relationship between 32x32 or 64x64 pixels and the one shown in Figure 2 (64px/256px)?
>
> **Re:** This might arise from misunderstanding. We are not visualizing one thorough diffusion process in Figure 2. In figure 2 we used a noise of fixed scale 0.3 to corrupt images of multiple resolutions, as mentioned in the caption below. What we want to express is that **the corruption by isotropic Gaussian noise of the same scale(0.3) to images of different resolutions have noticeably different effects**, especially in the low frequency part. However, **Block Noise of the same scale have similar effects on images of different resolutions** and can better destroy the low-frequency information of high-resolution images.
>
> > **Weakness 2:** The advantage of the proposed method is emphasized with training efficiency, however, it only considers the setting without CFG where some competitors do not yield the optimal output. I believe that it is better also more meaningful to compare all methods in the setting with CFG where the existing methods achieve the best performance (optimal setting). Furthermore, when claiming the efficiency, I would also expect the comparison of the inference time/steps of the proposed method with the existing ones in their best optimal regime (including sampling steps in Table 3.).
>
> **Re:** We actually want to make comparison of the intermediate results in the best regime but baselines including DiT[1], MDT[2], etc. only provide intermediate results without CFG (only report final results with CFG). When CFG is implemented, the performance of RDM in Figure 1 and other simlar comparisons will gain significant advancements as well.
>
> Also, we have added a column of sampling NFE for a clearer comparison in Table 3. The new table is presented as follows:
>
> https://i.postimg.cc/bwVyt8rh/new-table.jpg
>
> > **Weakness 3:** While the proposed method achieves slightly better sFID compared to MDT-XL/2-G on class-conditional generation ImageNet 256x256 (Table 3) (3.97 vs. 4.57, not really "a large margin" as stated in the abstract), however, most other metrics lag behind MDT-XL/2-G (such as FID, IS, and Recall). This indicates that to show its advantages over the existing approaches, it may need to present more evidence.
>
> **Re:** sFID is an improved version of FID, using spatial features instead of the standard pool features. It better captures spatial relationships and gives image distributions a coherent high-level structure. MDT[2] only drops sFID from previous SoTA 4.60 to 4.57, which indicates an improvement of 4.57 to 3.97 to be a large margin.
>
> From the modeling perspective, MDT has a complex structure 'asymmetric masking', making it tough to scale to higher resolutions and only perform well on 256x256 generation. In comparison, **RDM has better extendability and can easily scale to high-resolution image generation** without carefully adjusting the hyperparameters of noise schedule and architecture.
>
> To prove such a claim, we have conducted further experiments at higher resolutions (512 and 1024), training new RDM models on CelebA-HQ for 256-to-512 and 256-to-1024 super-resolution tasks. We tested their performance through a 3-stage generation process, reusing the previously released models for 64 and 64-to-256 resolutions. The RDM models achieved **SoTA performance on CelebA-HQ 512 and 1024 generation**, successfully demonstrating RDM's capability in high-resolution image generation. The results are presented as follows:
>
> https://i.postimg.cc/X71hwLST/celebahq-hres-results.jpg
>
> Additionally, we have included these results in Appendix C.1.

---

> ### Author Response · Authors · 2023-11-17
> **Response to Reviewer edjv (2/N)**
>
> > **Weakness 4:** Section 4.2 talking about CeleA-HQ, mentioned fewer training iterations while it is given with 50M and 820M trained images, making it a bit confusing that the dataset contained a hundred million images or that is just the total training steps. I recommend revising it more clearly and putting a column on the side of that table about the total training iterations or total training images.
>
> **Re:** For 'Number of Training Images', this term refers to **the total number of image samples that the model processes during its training phase, which is equal to (iters x batch_size)**.
>
> We appreciate your advice and have added a column of training cost, in the form of (iters x batch_size), in Table1 and Table3 for greater clarity. For example, the new Table 3 is presented as:
>
> https://i.postimg.cc/bwVyt8rh/new-table.jpg
>
> ======================== For Questions ========================
>
> > **Question 1:** What is the specific of each group in Table 3? It shows that the first column (Model) presents 5 groups (separated by \midrule) but unclear why is it divided like that.
>
> **Re:** In the 'Model' column of the table, we categorize the results based on different settings of model training and sampling strategies. The categories, listed from top to bottom, are:
>
> 1. GANs
> 2. Diffusion Models without CFG(Classifier-Free Guidance) used during sampling.
> 3. Cascaded Diffusion Models.
> 4. Diffusion Models with dynamic-scale CFG.
> 5. Diffusion Models with fixed-scale CFG.
>
> We hope this explanation clarifies how the models are organized in the table. To address any potential confusion regarding the table's presentation, we have restructured the divisions as follows:
>
> 1. GANs.
> 2. Diffusion Models without CFG.
> 3. Diffusion Models with CFG.
>
> The new table is presented as:
>
> https://i.postimg.cc/bwVyt8rh/new-table.jpg
>
> This revised table arrangement is now updated in Section 4.2 of our paper.
>
> > **Question 2:** I wondering if the number of data used in this paper is wrong, for example, Table 4 stated that 2500M training images were used for ImageNet64, is that 2.5 billion images? To my knowledge, ImageNet contains more than one million images, am I wrong? The same for all other datasets., does CelebA-HQ have 70 million images for training? Also, in Figure 1, is the horizontal axis the real number of images for training or it is just the iterations?
>
> **Re:** For 'Number of Training Images', this term refers to **the total number of image samples that the model processes during its training phase, which is equal to (iters x batch_size)**. This aligns with commonly reported metrics for training costs used by baselines such as DiT[1] and MDT[2].
>
> For our experiments, we used the original datasets from ImageNet and CelebA-HQ, which contain 1,281,167 samples and 30,000 samples, as elaborated in Section 4.1 of our paper. Similarly, in Figure 1, the horizontal axis illustrates the number of image samples that the model processes during training (iters x batch_size).
>
>
>
> In light of potential misunderstandings regarding the content of our paper, we welcome any further questions or clarifications you may need. Please feel free to reach out with any inquiries.
>
> Also, we believe that the demonstrated capability of RDM in high-resolution generation, **the achievement of SoTA on CelebA-HQ at 512 and 1024 resolutions**, marks a significant improvement.
>
> If the responses provided above have satisfactorily addressed your concerns, would you please consider increasing your rating accordingly?
>
>
>
> [1] Peebles, William, and Saining Xie. "Scalable diffusion models with transformers." *Proceedings of the IEEE/CVF International Conference on Computer Vision*. 2023.
>
> [2] Gao, Shanghua, et al. "Masked diffusion transformer is a strong image synthesizer." *arXiv preprint arXiv:2303.14389* (2023).

---

### Official Review · Reviewer_Jazo · 2023-10-29

**Soundness:** 3 good
**Presentation:** 4 excellent
**Contribution:** 3 good
**Rating:** 6
**Confidence:** 4

**Summary:**

This paper focuses on the task of using a diffusion model for high-resolution image generation. Specifically, the authors locate the issue in the SNR in the frequency domain and introduce the block noise to bridge the gap and build an RDM upon a cascaded pipeline to get rid of the reliance on the low-resolution condition. The authors show state-of-the-art FID on CelebA-HQ and ImageNet on 256x256.

**Strengths:**

1. this paper has excellent presentation quality and it is easy for readers to follow.

2. this paper provides a comprehensive analysis of the limitation of existing cascaded model for high-resolution image generation and locate the issue in the SNR of higher-resolution image. To address these issues, the authors introduce RDM to solve this issue. I think this highlight is insightful and could inspire others in the community.

3. the authors have shown extensive comparison in the paper to verify the effectiveness of the RDM.

**Weaknesses:**

1. This paper focuses on high-resolution image generation. However, the biggest image resolution used in this paper is only 256x256, which is much smaller than the existing definition of ‘high-resolution‘. I would expect an experiment result that has a resolution at least 512 or 1024 to see whether RDM still works.

2. The authors claim that any artifacts in the low-res images can be corrected in the high-res stage, we expect some qualitative cases in experiments to verify such a claim.

**Questions:**

This paper is well-written and the proposal shows some impressive results in experiments. My major concerns are about the experiment on really high-resolution images and some experiments to verify the superiority of the RDM further. Please see more details in the Weaknesses.

---

> ### Author Response · Authors · 2023-11-16
> **Response to Reviewer Jazo (1/N)**
>
> Thank you for your valuable comments, we will explain your concerns point by point.
>
> ======================== For Weakness ========================
>
> > **Weakness 1:** This paper focuses on high-resolution image generation. However, the biggest image resolution used in this paper is only 256x256, which is much smaller than the existing definition of ‘high-resolution‘. I would expect an experiment result that has a resolution at least 512 or 1024 to see whether RDM still works.
>
> **Re:** We have conducted further experiments at higher resolutions (512 and 1024), training new RDM models on CelebA-HQ for 256-to-512 and 256-to-1024 super-resolution tasks. We tested their performance through a 3-stage generation process, reusing the previously released models for 64 and 64-to-256 resolutions. The RDM models achieved **SoTA performance on CelebA-HQ 512 and 1024 generation**, successfully demonstrating RDM's capability in high-resolution image generation. The results are presented as follows:
>
> https://i.postimg.cc/X71hwLST/celebahq-hres-results.jpg
>
> Additionally, we have included these results in Appendix C.1.
>
> > **Weakness 2:** The authors claim that any artifacts in the low-res images can be corrected in the high-res stage, we expect some qualitative cases in experiments to verify such a claim.
>
> **Re:** To demonstrate the capability of RDM to correct artifacts from the low-resolution stage, we conducted additional experiments comparing super-resolution generation between RDM and ADM-U. We used 64x64 samples from ImageNet as low-resolution inputs, and introduced  artifacts by adding 0.1 scale of Gaussian Noise. RDM successfully managed the noise, producing clean 256x256 samples, while ADM-U transferred the noise to its 256x256 samples. The comparison is presented as follows:
>
> https://i.postimg.cc/Nj1pyVh5/artifacts.jpg
>
> We have also included this comparison in our paper under Appendix C.3.
>
> Additionally, recognizing the potential for overstatement, we have amended our initial claim from 'any artifacts can be corrected' to 'artifacts could be corrected' for a more accurate representation.
>
> ======================== For Questions ========================
>
> > **Question 1:** This paper is well-written and the proposal shows some impressive results in experiments. My major concerns are about the experiment on really high-resolution images and some experiments to verify the superiority of the RDM further. Please see more details in the Weaknesses.
>
> **Re:** Thank you very much for your recognition! Please refer to our reply for **Weakness 1** and **Weakness 2** to check our additional experiments on further verification of RDM's capability.
>
>
>
> From our perspective, the demonstration of RDM's capability in high-resolution generation, achieving **SoTA results on CelebA-HQ at 512 and 1024 resolutions**, represents a significant improvement.
>
> If the replies provided above have addressed your concerns, would you please consider revising your rating upwards?

---

### Official Review · Reviewer_htaY · 2023-10-30

**Soundness:** 2 fair
**Presentation:** 4 excellent
**Contribution:** 3 good
**Rating:** 8
**Confidence:** 4

**Summary:**

This work designs a diffusion framework for medium-resolution image generation and explores it on unconditional CelebA-HQ $256^2$ and class-conditional ImageNet $256^2$. The framework generally follows EDM and cascaded diffusion model and develops some novel contributions on top of them. First, it develops a novel diffusion/denoising process formulation, where the noise, added to an image, is locally correlated for the higher resolution images. Then, instead of concatenating the upsampled low-resolution images channel-wise, the diffusion process is unified for low and higher resolution images. As such, it does not need to train the low-resolution generator from scratch and uses the pre-trained EDM checkpoints. The framework achieves SOTA FID results on ImageNet $256^2$ and CelebA-HQ $256^2$. The paper also contains interesting analysis of the frequencies of noised/clean images in low/medium resolutions and also some curious insights about the cascaded diffusion models.

**Strengths:**

- The work sets a new state-of-the-art results on ImageNet $256^2$ for constant CFG.
- Analysing SNRs across frequencies is interesting and might inspire some subsequent works to explore this direction as well.
- The work has not only proposed the idea of block-wise locally correlated noise, but also derived the sampler for it (while the derivations are not particularly involved, they still require some carefulness).
- The exposition is very clear and the paper is easy to follow.

**Weaknesses:**

- $256^2$ resolution is not "high-resolution" (as claimed in the paper), but rather "medium-resolution". Even the $512^2$ resolution can be fit with end-to-end architectures — e.g. simple diffusion [1] or VDM++ [2] (they can even fit 512x512 in the end-to-end fashion). In this way, it's no clear whether the method would easily scale for high-resolution generation without re-tuning the hyperparameters.
- Scaling Laws clearly demonstrated that the amount of compute spent on training the model has great influence on the final performance — just like the developed novel techniques or model size. However, there is no information about the training cost in the paper (only in terms of the amount of iterations). The paper mentioned that the $256^2$-resolution generator is ~10x more expensive that the $64^2$-resolution one, so I would guess that it might be way more expensive in terms of compute compared to the baselines.
- Figure 1 (and the accompanying claims about faster convergence in the text) seems misleading: the model is not trained from scratch, but compares the convergence with from-scratch trained models. The vanilla EDM was trained on
- class-balanced FID is the known trick to improve the performance, and, since it's unclear whether prior works used it as well, shouldn't be used in the claims about SotA FID (having it in the table as a separate row is not an issue).
- f-DM (Gu et ICLR'22) can also be seen as a "relay" diffusion, but it is not even mentioned in the paper.

[1] simple diffusion: End-to-end diffusion for high resolution images
[2] Understanding Diffusion Objectives as the ELBO with Simple Data Augmentation
[3] f-DM: A Multi-stage Diffusion Model via Progressive Signal Transformation

**Questions:**

- Would the method work on $512^2$, $1024^2$ and/or higher resolutions?
- How does the method perform with from-scratch training? If the model cannot train well from scratch, then the claim about its simplicity compared to CDMs does not hold and should be removed. Also, it's not fair to claim the reduced amount of training steps for a non-from-scratch trained model.
- Can it be the case that the reduced amount of sampling steps is due to the 2-nd order sampler? For example, MDT uses DDPM sampler, which was shown by EDM to require more sampling steps.
- For a fair comparison, please report the training/inference costs of the developed model vs the baselines (where possible).
- Please, update the claims about the convergence speed (especially the teaser paper) to specify that the model is not trained from scratch.
- Please, update the claims about SotA FID based on the unbalanced FID instead of the class-balanced FID.
- Why not try dynamic CFG as well to further improve FIDs? Or it was tried and didn't work?
- It would be good to describe the differences with f-DM.

---

> ### Author Response · Authors · 2023-11-17
> **Response to Reviewer htaY (1/N)**
>
> Thank you for your valuable comments, we will explain your concerns point by point. Our response is divided into 3 parts.
>
> ======================== For Weakness ========================
>
> > **Weakness 1:** 256^2 resolution is not "high-resolution" (as claimed in the paper), but rather "medium-resolution". Even the 512^2 resolution can be fit with end-to-end architectures — e.g. simple diffusion [1] or VDM++ [2] (they can even fit 512x512 in the end-to-end fashion). In this way, it's no clear whether the method would easily scale for high-resolution generation without re-tuning the hyperparameters.
>
> **Re:** We have conducted further experiments at higher resolutions (512 and 1024), training new RDM models on CelebA-HQ for 256-to-512 and 256-to-1024 super-resolution tasks. We tested their performance through a 3-stage generation process, reusing the previously released models for 64 and 64-to-256 resolutions. The RDM models achieved **SoTA performance on CelebA-HQ 512 and 1024 generation**, successfully demonstrating RDM's capability in high-resolution image generation. The results are presented as follows:
>
> https://i.postimg.cc/X71hwLST/celebahq-hres-results.jpg
>
> Additionally, we have included these results in Appendix C.1.
>
> > **Weakness 2:** Scaling Laws clearly demonstrated that the amount of compute spent on training the model has great influence on the final performance — just like the developed novel techniques or model size. However, there is no information about the training cost in the paper (only in terms of the amount of iterations). The paper mentioned that the 256^2 resolution generator is ~10x more expensive that the 64^2 resolution one, so I would guess that it might be way more expensive in terms of compute compared to the baselines.
>
> **Re:** This is a misunderstanding. We have already compared the training costs of RDM with other baselines, which are also 256x256 models. For RDM, we calculate the training cost based on the number of training samples (iters x batch size) on 256 resolution. This method aligns with commonly reported metrics for training costs used by baselines such as DiT[1] and MDT[2].
>
> To the best of our knowledge, **(FLOPS x iters x batch_size) would be a precise measure of training cost**. Since we borrow the UNet structure from ADM[3], which ensures a consistent FLOPS count, we consider (iters x batch_size) to be a fair standard.
>
> Additionally, **we have included two stages in our calculation of the training cost**. As explained in Appendix B, the FLOPS of the 64-resolution model are less than 1/10 of the 256-resolution model. So we have incorporated the training cost of the low-resolution stage as 1/10 into the total cost. For greater clarity, we have now included the training cost calculation process in the 'Architecture and Training' section of Section 4.1.
>
> > **Weakness 3:** Figure 1 (and the accompanying claims about faster convergence in the text) seems misleading: the model is not trained from scratch, but compares the convergence with from-scratch trained models. The vanilla EDM was trained on
>
> **Re:** This is a misunderstanding. The models can train well from scratch.
>
> We guess the misunderstanding is from 'we directly use the released checkpoint from EDM in ImageNet in the 64 × 64 stage'. However, **the 64-to-256 model is a separate model from the first stage model** (64x64 model) and **both stages are trained from scratch**.
>
> EDM[4] released its code, checkpoints as well as all the training hyperparameters for 64x64 models. This makes training the first stage of RDM on ImageNet for 64x64 generation a reproducible and repetitive work from EDM. We have verified the reproducibility of EDM by repeat the early training of EDM's 64x64 ImageNet model. Yet eventually, to avoid wasting computing resources, we directly borrow the pretrained EDM 64x64 ImageNet model as the first stage of RDM on ImageNet. In our calculation of total training consumption, the training cost of the pretrained model is also counted.
>
> While in the experiment of CelebA-HQ, both stages of models are trained from scratch by ourselves.
>
> > **Weakness 4:** class-balanced FID is the known trick to improve the performance, and, since it's unclear whether prior works used it as well, shouldn't be used in the claims about SotA FID (having it in the table as a separate row is not an issue).
>
> **Re:** We only claim the best sFID but not FID on ImageNet 256. Also, without class-balancing, RDM still achieves SoTA sFID on ImageNet 256x256, and the second place ranking on FID (inferior to MDT). sFID is an improved version of FID, using spatial features instead of the standard pool features. It better captures spatial relationships and gives image distributions a coherent high-level structure.

---

> ### Author Response · Authors · 2023-11-17
> **Response to Reviewer htaY (2/N)**
>
> > **Weakness 5:** f-DM (Gu et ICLR'22) can also be seen as a "relay" diffusion, but it is not even mentioned in the paper.
>
> **Re:** There are distinguishable aspects of difference between them as follows:
>
> - Compare to f-DM using linear interpolation, we model the relaying resolutions with **Heat Dissipation (Blurring process)**.
> - We introduce **Block Noise** to further bridge the gap between stages and corrupt low-frequency information, which is an essential part of RDM and brings significant improvement.
> - While f-DM uses one model for unified multiple stages’ generation, RDM uses **separate models to handle different stages**. Encoding multiple resolutions of generation into one model could be difficult for a UNet model to learn. In comparison, RDM breaks down the generation task into multiple, simpler tasks, which largely ease the training.
> - On ImageNet 256x256, **the performance of RDM is far improved** compared to f-DM.
> - RDM shows **greater extendability over f-DM**. For example, if the generation on higher resolution is desired, by RDM we can simply train a new super-resolution model on relaying and reuse the previous stages (exactly what we do to extend to CelebA-HQ 512 and 1024 generation). While for f-DM, the arrangement of the schedule and the training of the model, etc., are required to be conducted all over again.
>
> Thank you very much for pointing out. f-DM and RDM share the idea to unify multiple resolutions of image generation in one diffusion schedule, which is a work we didn't previously draw attention to. We will definitely add the mention of f-DM and discussion of the difference to our paper later.
>
> ======================== For Questions ========================
>
> >  **Question 1:** Would the method work on 512^2 1024^2 and/or higher resolutions?
>
> **Re:** Please refer to our reply for **Weakness 1**. We have further verify the capability of RDM on high-resolution generation, achieving **SoTA on CelebA-HQ 512 and 1024**.
>
> > **Question 2:** How does the method perform with from-scratch training? If the model cannot train well from scratch, then the claim about its simplicity compared to CDMs does not hold and should be removed. Also, it's not fair to claim the reduced amount of training steps for a non-from-scratch trained model.
>
> **Re:** This is a misunderstanding. The models can train well from scratch.
>
> In RDM, **the 64-to-256 model is a separate model from the first stage model** (64x64 model) and **both stages are trained from scratch**.
>
> we have calculate the training cost based on the number of training samples (iters x batch_size) on 256 resolution and compared with other baselines. This method aligns with commonly reported metrics for training costs used by baselines such as DiT[1] and MDT[2]. To make a fair comparison, **we have included two stages of RDM in our calculation of the training cost**. As explained in Appendix B, the FLOPS of the 64-resolution model are less than 1/10th of the 256-resolution model. So we have incorporated the training cost of the low-resolution stage as 1/10th into the total cost. For greater clarity, we have now included the training cost calculation process in the 'Architecture and Training' section of Section 4.1.
>
> > **Question 3:** Can it be the case that the reduced amount of sampling steps is due to the 2-nd order sampler? For example, MDT uses DDPM sampler, which was shown by EDM to require more sampling steps.
>
> We supplement the experiments on the first-order sampler. All models use first-order sampler for generation on ImageNet-256 with 80 sampling steps. RDM still outperforms MDT and DiT. The result is as follow:
>
> https://i.postimg.cc/hP6cMYCV/ddim-comparison.png

---

> ### Author Response · Authors · 2023-11-17
> **Response to Reviewer htaY (3/N)**
>
> > **Question 4:** For a fair comparison, please report the training/inference costs of the developed model vs the baselines (where possible).
>
> **Re:** We have added training cost in Table 1 and Table 3. The new Table 3 is presented as follows, for example:
>
> https://i.postimg.cc/bwVyt8rh/new-table.jpg
>
> Please refer to the reply for **Weakness 2** for a detailed interpretation of the calculation of the training cost.
>
> > **Question 5:** Please, update the claims about the convergence speed (especially the teaser paper) to specify that the model is not trained from scratch.
>
> **Re:** This is a misunderstanding as the RDM models are trained from scratch. Please refer to the reply for **Weakness 3** for a detailed explanation.
>
> > **Question 6:** Please, update the claims about SotA FID based on the unbalanced FID instead of the class-balanced FID.
>
> **Re:** We only claim the best sFID but not FID on ImageNet 256. Also, without class-balancing, RDM still achieves SoTA sFID on ImageNet 256x256, and the second place ranking on FID (inferior to MDT). sFID is an improved version of FID, using spatial features instead of the standard pool features. It better captures spatial relationships and gives image distributions a coherent high-level structure.
>
> > **Question 7:** Why not try dynamic CFG as well to further improve FIDs? Or it was tried and didn't work?
>
> **Re:** In the experiment of ImageNet 256x256, the low resolution stage of RDM (64x64) is trained without condition dropout,. Therefore the low-resolution stage of RDM is not able to implement dynamic CFG (or fixed CFG), while the improvement of only implementing CFG on the second stage is much restricted.
>
> > **Question 8:** It would be good to describe the differences with f-DM.
>
> **Re:** There are distinguishable aspects of difference between them. Please refer to our reply for **Weakness 5** for a detailed description of the differences.
>
> Thank you very much for pointing out. f-DM and RDM share the idea to unify multiple resolutions of image generation in one diffusion schedule, which is a work we didn't previously draw attention to. We will definitely add the mention of f-DM and discussion of the difference to our paper later.
>
>
>
> In light of potential misunderstandings regarding the content of our paper, we welcome any further questions or clarifications you may need. Please feel free to reach out with any inquiries.
>
> Also, we believe that the demonstrated capability of RDM in high-resolution generation, **the achievement of SoTA on CelebA-HQ at 512 and 1024 resolutions**, marks a significant improvement.
>
> If the responses provided above have satisfactorily addressed your concerns, would you please consider increase your rating accordingly?
>
> [1] Peebles, William, and Saining Xie. "Scalable diffusion models with transformers." Proceedings of the IEEE/CVF International Conference on Computer Vision. 2023.
>
> [2] Gao, Shanghua, et al. "Masked diffusion transformer is a strong image synthesizer." arXiv preprint arXiv:2303.14389 (2023).
>
> [3] Dhariwal P, Nichol A. Diffusion models beat gans on image synthesis[J]. Advances in neural information processing systems, 2021, 34: 8780-8794.
>
> [4] Karras T, Aittala M, Aila T, et al. Elucidating the design space of diffusion-based generative models[J]. Advances in Neural Information Processing Systems, 2022, 35: 26565-26577.

---

> ### Comment · Reviewer_htaY · 2023-11-22
> **Reviewer's response to the rebuttal (part 1/2)**
>
> Thank you for providing the response to my feedback, it really helped me to understand your work better. Let me share my thoughts:
>
> - **Weakness 1 (about the high-res generation)**.
> Thank you for providing these results, I find them compelling and updated my score accordingly to reflect that.
> One concern though is that the models are not entirely comparable in the provided table in terms of FID since they have different amount of parameters (your cumulative model is ~2B parameters which is huge for CelebA) and training costs, but I am not using this concern to affect my rating since it is nearly impossible to equalize/factor-out all the differences between the existing methods.
>
> -  **Weakness 2 (about the training cost)**.
> > To the best of our knowledge, (FLOPS x iters x batch_size) would be a precise measure of training cost
>
> I respectfully disagree with that, and believe that hiding the wall-clock-time training costs behind flops/iters/etc. is misleading. RepVGG is a reasonable reference which highlights this issue within the community: https://arxiv.org/abs/2101.03697. The real training cost should be measured as the amount of GPU-hours/days/etc (depending on the scale) — and EDM, which the current work builds upon, is doing precisely that: please take a look at their table in the README in https://github.com/NVlabs/edm. Also, FLOPS or MACs are not a good way to measure the speed, since modern DL models are memory-bound and compute-bound.
>
> - **Weakness 3 (about Fig 1 displaying from-scratch vs non-from-scratch trained models)**
> Apparently, my wording was not entirely clear. Do I get it right that in figure 1, you compare the convergence of the 64->256 upsampler vs the convergence of the from-scratch-trained generators? If so, this is for sure misleading.
>
> - **Weakness 4 (about tweaking FID with class-balancing).**
> > We only claim the best sFID but not FID on ImageNet 256.
>
> In Fig 1, you state "RDM can achieve an FID of 1.87". You could have stated something like "RDM can achieve a class-balanced FID (FID-CB) of 1.87". Let's distinguish FID and FID-CB, since otherwise, someone who passes through the paper can think that you achieved the traditional FID of 1.87 if they do not look deep into the paper.
> > RDM still achieves SoTA sFID [...] sFID is an improved version of FID
>
> Could you please point out to some evidence that sFID is indeed an improved version of FID (e.g., some human study showing that it correlates better with the perceived image quality)? Otherwise, it's difficult to take such a claim.

---

> > ### Comment · Reviewer_htaY · 2023-11-22
> > **Reviewer's response to the rebuttal (part 2/2)**
> >
> > - **Weakness 5 (about the comparison with f-DM**. Thank you for highlighting the differences with f-DM, it really helped me to understand your contributions better.
> > - **Question 1 (about higher resolutions)**. The newly provided results fully resolve my concern.
> > - **Question 2 (about from-scratch training)**. I apologize, my wording was not precise enough: what I meant is end-to-end training, when we train all the stages jointly.
> > - **Question 3 (about second-order sampling)**. I would withdraw my concern here, since I guess it's impossible to reliable compare sampling speeds of the methods trained with different diffusion parametrizations. But I am grateful to the authors for providing additional evaluations with the first-order sampling.
> > - **Question 4 (about training/inference costs)**. Here, my concern remains fully — please see #2 above.
> > - **Question 5 (about the convergence speed)**. Here, my concern remains fully — please see #3 above.
> > - **Question 6 (claims about FID)**. For me, FID $\neq$ FID-CB (class-balanced FID), but you use them interchangeably throughout your work (e.g., Fig 1), and this can mislead a reader. Frankly, it's unclear what is the best way to proceed here, since inventing one more new metric (FID-CB) does not seem like the best idea as well.
> > - **Question 7 (about dynamic CFG)**. The provided response fully resolved my concern: I understand that it would be difficult to experiment with it in your case. But I guess that the base stage could be easily fine-tuned with label dropout (please note that it's not a request for additional experiments from my side, just a thought which might be helpful to you).
> > - **Question 8 (about the comparison with f-DM)**. Here, my concern is fully resolved, thank you for providing these details.
> >
> > **To sum up**:
> > - My concerns W1, W5, Q1, Q3, Q7, Q8 have been fully resolved, but W2, W3, W4, Q2, Q4, Q5, Q6 still remain. I've updated my rating accordingly. My intention is to give the rating of 7, but there was no such rating available in the console, which is why I set 8 since setting 6 felt too unfair given the contributions. But I've communicated to the AC that my current 8 is actually 7.
> > - My current perspective of this work is that it is an improved version of cascaded DMs with a better noise schedule. I would be curious to know if you feel that my perception is incorrect and I am missing some important aspects.
> > - My main concern right now is that the paper overclaims several its results (class-balanced FID, training cost, convergence speed). It would be a good contribution otherwise if the results are presented as they are.

---

> ### Author Response · Authors · 2023-11-23
> **Response to Reviewer htaY (4/N)**
>
> Thank you again for your kind and valuable response. For your remaining concerns, we have made further explanations and revisions as follows, which we hope would help the whole discussion:
>
> > **Weakness 2 (about the training cost)**.
> >
> > > To the best of our knowledge, (FLOPS x iters x batch_size) would be a precise measure of training cost
> >
> > I respectfully disagree with that, and believe that hiding the wall-clock-time training costs behind flops/iters/etc. is misleading. RepVGG is a reasonable reference which highlights this issue within the community: https://arxiv.org/abs/2101.03697. The real training cost should be measured as the amount of GPU-hours/days/etc (depending on the scale) — and EDM, which the current work builds upon, is doing precisely that: please take a look at their table in the README in https://github.com/NVlabs/edm. Also, FLOPS or MACs are not a good way to measure the speed, since modern DL models are memory-bound and compute-bound.
> >
> > **Question 4 (about training/inference costs)**. Here, my concern remains fully — please see #2 above.
>
> **Re:** While previous SoTA models DiT[1] and MDT[2] both only report the training cost by (iter x batch_size), therefore we formerly chose to do so as well to make a fair comparison possible. We respect your opinions on the measurement of the training cost and, to make a more complete statement, we have now measured our training cost by GPU hours/days as follows:
>
> -  On ImageNet
>    - The first stage model (64 generation) was trained on 32 V100 for 13 days (according to EDM).
>    - The second stage model (64-256  generation) was trained on 64 A100-40G for 12.5 days.
> -  On CelebA-HQ
>    - The first stage model (64 generation) on 32 A100-40G for 16 hours.
>    - The second stage model (64-256  generation) on 32 40G-A100 for 25.5 hours.
>
> We have added these statistics to Appendix B.2.
>
>
>
> > **Weakness 3 (about Fig 1 displaying from-scratch vs non-from-scratch trained models)** Apparently, my wording was not entirely clear. Do I get it right that in figure 1, you compare the convergence of the 64->256 upsampler vs the convergence of the from-scratch-trained generators? If so, this is for sure misleading.
> >
> > **Question 2 (about from-scratch training)**. I apologize, my wording was not precise enough: what I meant is end-to-end training, when we train all the stages jointly.
> >
> > **Question 5 (about the convergence speed)**. Here, my concern remains fully — please see #3 above.
>
> **Re:** When plotting the curve of RDM in Figure 1, **we have added the training cost of the first stage (as 1/10 according to their flops) to get a total training consumption for RDM's place on the x-axis**, as we previously stated. Therefore, we don't assume we are comparing the convergence of the upsampler versus end-to-end models. But it is true that we are comparing the convergence of **a whole cascaded pipeline versus end-to-end models**. A cascaded pipeline trains the model of each stage independently and cannot train all stages jointly. Perceiving that a cascaded pipeline may contribute to a faster convergence, we also have ADM-U (a previous cascaded setting) on Figure 1, whose performance lags behind RDM. This further proves the superiority of RDM on better convergence.
>
> It may be less correlated with this specific issue, but we have also tested some of the components from RDM on end-to-end diffusion training. We have tried adding Block Noise to 256x256 end-to-end training, which exhibits positive results. We trained ADM on CelebA-HQ 256x256 w/ and w/o Block Noise and tested them on FID-5K, finding that **ADM with block noise achieves better and more stable FID performance, which verifies the effectiveness of the block noise on end-to-end models**. The result is as follows:
>
> https://i.postimg.cc/GhH5mbjQ/block-noise-end2end.jpg
>
> We assume this result may have further connections with the addition of offset-noise in SDXL[3], etc.

---

> > ### Author Response · Authors · 2023-11-23
> > **Response to Reviewer htaY (5/N)**
> >
> > > **Weakness 4 (about tweaking FID with class-balancing).**
> > >
> > > > We only claim the best sFID but not FID on ImageNet 256.
> > >
> > > In Fig 1, you state "RDM can achieve an FID of 1.87". You could have stated something like "RDM can achieve a class-balanced FID (FID-CB) of 1.87". Let's distinguish FID and FID-CB, since otherwise, someone who passes through the paper can think that you achieved the traditional FID of 1.87 if they do not look deep into the paper.
> > >
> > > > RDM still achieves SoTA sFID [...] sFID is an improved version of FID
> > >
> > > Could you please point out to some evidence that sFID is indeed an improved version of FID (e.g., some human study showing that it correlates better with the perceived image quality)? Otherwise, it's difficult to take such a claim.
> > >
> > > **Question 6 (claims about FID)**. For me, FID ≠ FID-CB (class-balanced FID), but you use them interchangeably throughout your work (e.g., Fig 1), and this can mislead a reader. Frankly, it's unclear what is the best way to proceed here, since inventing one more new metric (FID-CB) does not seem like the best idea as well.
> >
> > **Re:** To completely remove the possible misleadingness, we have **updated our original claim from 'RDM can achieve an FID of 1.87' to 'RDM can achieve a FID of 1.99 (and a class-balanced FID of 1.87)'**, where 1.99 is the FID achieved without CB. While the claimed FID dropped by a certain amount, it is worth noticing that by 1.99 RDM still **maintains a second place ranking on the leaderboard of ImageNet 256 generation** (inferior to 1.79 by MDT[2], but better than 2.27 by DiT[1]).
> >
> > As an updated version of FID, sFID uses spatial features rather than the standard pooled features. The sFID is proposed by paper *Generating images with sparse representations*, which states that this metric better captures spatial relationships, rewarding image distributions with coherent high-level structure. However, further studies for the correlation of these metrics and perceptual quality are still absent. At least we believe testing on sFID could **help to extend the scope of evaluation** and prevent works of generative models from only overfitting on traditional FID.
> >
> >
> >
> > Thanks again for your valuable response. Hope the updated replies could further address your concerns.
> >
> >
> >
> > [1] Peebles, William, and Saining Xie. "Scalable diffusion models with transformers." *Proceedings of the IEEE/CVF International Conference on Computer Vision*. 2023.
> >
> > [2] Gao, Shanghua, et al. "Masked diffusion transformer is a strong image synthesizer." *arXiv preprint arXiv:2303.14389* (2023).
> >
> > [3] Podell, Dustin, et al. "Sdxl: Improving latent diffusion models for high-resolution image synthesis." *arXiv preprint arXiv:2307.01952* (2023).

---

### Official Review · Reviewer_pR8K · 2023-11-01

**Soundness:** 3 good
**Presentation:** 3 good
**Contribution:** 3 good
**Rating:** 8
**Confidence:** 3

**Summary:**

This paper presents the Relay Diffusion Model (RDM), a new cascaded framework to improve the shortcomings of the previous cascaded methods. Contributions are (1) The difference of noise addition process between low-resolution and high-resolution image diffusion is analyzed from the perspective of frequency domain. Based on the analyzation, this paper further introduces the block noise to bridge the gap; (2)  By combing a low-resolution ordinary diffusion model and a high-resolution blurring diffusion model, RDM starts diffusion from the low-resolution result instead of pure noise, reducing the training and sampling steps; and (3) This paper also evaluates the effectiveness of RDM on unconditional CelebA-HQ 256×256 and conditional ImageNet 256×256 datasets.

**Strengths:**

1. For originality, this paper first combines a low-resolution ordinary diffusion model with a high-resolution blurring diffusion model. To some extent, this solves the problem of noise schedule in high-resolution diffusion. Besides, the analysis of the noise gap between low-resolution and high-resolution is interesting.
2. For clarity, this paper’s writing is well-structured, and the problem was presented and solved straightforwardly.

**Weaknesses:**

1. The transition from section 3.1 to section 3.2 is a little incomprehensible; why combining block noise and blurring diffusion model in the second stage needs further explanation. Besides, from the equation and algorithm, I can’t find an explicit connection between low-resolution diffusion and high-resolution diffusion; such an explicit statement may be necessary for reader to understand.
2. Missing experiments: How does an end-to-end model for high-resolution images by introducing block noise in early diffusion steps perform? The author mentions it on page 5 but does not explain it.

**Questions:**

1. On page 2, “Training Efficiency”, the author takes the cascaded method as a solution to mitigate memory and computation costs, but the cascaded method still needs to train and inference in the highest resolution.  How does such a setting mitigate memory and computation cost? For RDM, the problem still exists.
2. I can’t understand why, for Fig.2, the high-frequency period is meaningless, and the difference can be neglected. More explanation may be required.
3. How to choose $D_{T}^{p}$ to guarantee $VD_{T}^{P}V^{T}x_{0}$ in the same distribution as $x^{H}$; this is the key to connect low-resolution diffusion and high-resolution diffusion.
4. Adding the training iterations of each method in Tables 1, and 3 may make it more readable. The comparison may be a little unfair because RDM’s low-resolution diffusion model is pre-trained.
5. For the text description below equation (11), why blurring corruption and block noise corruption can be considered independently, i.e., why can we just replace $\epsilon$ with $\tilde{\epsilon}$? Is there any formal proof?
6. For Algorithm 1 in A.4, it seems that the sampling algorithm does not contain the process of the low-frequency diffusion model. Please clarify.
7. Some minor errors:
    * On page 4, two “the” in “the same noise level on a higher resolution results in a higher SNR in the (low-frequency part of) the frequency domain.”
    *  For equation (6), the denominator should be $s$ or $s^2$?
    *  Equation 8 may be not appropriate, the $\epsilon^{'}$ should be below the expectation symbol too.


-----------
Thank the authors for their responses. Most of my concerns have been addressed. I'd like to raise my score.

---

> ### Author Response · Authors · 2023-11-16
> **Response to Reviewer pR8K (1/N)**
>
> Thank you for your valuable comments, we will explain your concerns point by point. Our response is divided into 3 parts.
>
> ======================== For Weakness ========================
>
> > **Weakness 1:** The transition from section 3.1 to section 3.2 is a little incomprehensible; why combining block noise and blurring diffusion model in the second stage needs further explanation. Besides, from the equation and algorithm, I can’t find an explicit connection between low-resolution diffusion and high-resolution diffusion; such an explicit statement may be necessary for reader to understand.
>
> **Re:** Please refer to the replies for **Question 3** and **Question 6**, which explained the connection of low-res and high-res stages. The incomprehensibility may be largely from the error in Algorithm 1, where the starting point of sampler is supposed to be '$X_N$ ~ low-resolution images' and we have corrected it in Algorithm 1. From the perspective of forward process, the final state of a patch-wise blurring process is a nearest-upsampled version of a low-resolution image. This guarantees a seamless integration between different resolutions.
>
> We have explained the combination of block noise and blurring diffusion in the reply for **Question 5**. The blurring process interpolates between an upsampled low-resolution start and a high-resolution end, while the noise corruption meantime performs a standard diffusion process.
>
> > **Weaknesses 2:** Missing experiments: How does an end-to-end model for high-resolution images by introducing block noise in early diffusion steps perform? The author mentions it on page 5 but does not explain it.
>
> **Re:** We have conducted further ablation study of block noise on an end-to-end model and added it in Appendix C.2. We trained ADM on CelebA-HQ 256x256 w/ and w/o block noise and tested them on FID-5K. The result is as follows:
>
> https://i.postimg.cc/GhH5mbjQ/block-noise-end2end.jpg
>
> We found that ADM with block noise achieves better and more stable FID performance, which verifies the effectiveness of the block noise on end-to-end models.

---

> > ### Author Response · Authors · 2023-11-16
> > **Response to Reviewer pR8K (2/N)**
> >
> > ======================== For Questions ========================
> >
> > > **Question 1:** On page 2, “Training Efficiency”, the author takes the cascaded method as a solution to mitigate memory and computation costs, but the cascaded method still needs to train and inference in the highest resolution. How does such a setting mitigate memory and computation cost? For RDM, the problem still exists.
> >
> > **Re:** Compared to end-to-end high-resolution training, a cascaded pipeline breaks down the generation task into multiple, simpler tasks across increasing resolutions. This approach significantly reduces the burden of training diffusion models at high resolutions. Although training at lower resolutions is still necessary, it requires far less computational resources, as evidenced by the reduced FLOPS. For example, In RDM, training for 64x64 resolutions needs only 1/10 FLOPS compared to 256x256 models.
> >
> > This efficiency is a key reason why cascaded configurations have become the standard in current SoTA text-to-image models, such as DALL-E[1] and Imagen[2], as they help to 'mitigate memory and computation costs'.
> >
> > Furthermore, RDM is even more efficient than typical cascaded pipelines. In RDM, the high-resolution stage begins diffusion from a midpoint in the schedule, cutting more than half of the sampling steps compared to conventional cascaded models. In contrast, these models typically generate through an entire diffusion schedule at the high-resolution stage.
> >
> > > **Question 2:** I can’t understand why, for Fig.2, the high-frequency period is meaningless, and the difference can be neglected. More explanation may be required.
> >
> > **Re:** In Figure 2(a), the high-frequency component is meaningless because, to make a meaningful comparison in the same frequency space, we first upsample the 64x64 image to a 256x256 image. Therefore, in the Signal-to-Noise Ratio (SNR) curve, the information in the high-frequency part is a product of the upsampling-and-interpolation process and holds little significance. However, the high-frequency part in Figure 2(c) is meaningful. The block noise in this figure does not significantly corrupt the high-frequency information. Consequently, we opted for a mixture of Gaussian noise and block noise, which more effectively corrupts both the low-frequency and high-frequency information in the image.
> >
> > > **Question 3:** How to choose $D^P_T$ to guarantee $VD^P_TV^Tx_0$ in the same distribution as $x^H$; this is the key to connect low-resolution diffusion and high-resolution diffusion.
> >
> > **Re:** We provide the detailed derivation of $D^P_T$ in Appendix A.1. $D^P_T$ is defined as a patch-wise blurring transformation matrix. In a patch-wise blurring process, the final state of the image results in each patch having a uniform pixel value, which is the average value of the patch at the beginning. For a 4x4 patch in the RDM setting, this approach ensures that the final state of the patch-wise blurring will be a nearest-upsampled version of a low-resolution image. This guarantees a seamless integration between different resolutions.

---

> ### Author Response · Authors · 2023-11-16
> **Response to Reviewer pR8K (3/N)**
>
> > **Question 4:** Adding the training iterations of each method in Tables 1, and 3 may make it more readable. The comparison may be a little unfair because RDM’s low-resolution diffusion model is pre-trained.
>
> **Re:** The claim of the unfair comparison may be a misunderstanding. We have factored in two stages when calculating the training cost. We calculate the training cost as the number of training samples (iters x batch_size) on 256 resolution. As mentioned in Appendix B, the FLOPS of the 64-resolution model are less than 1/10 of the 256-resolution model, so we have incorporated the training cost of the 64-resolution stage as 1/10 to the total cost. Same incorporation is also conducted in the ablated comparison of sampling steps in Section 4.3. For greater clarity, we have also detailed the process of calculating the training cost in the 'Architecture and Training' section, found in Section 4.1.
>
> Thank you for the suggestion. In response to considerations about the different training batch sizes used in previous works, we have revised our paper to add a column titled 'Number of Training Samples (iter x batch_size)' in the table as follows.
>
> https://i.postimg.cc/L6BLG8hX/table3-new.jpg
>
> We hope this will make a more clear comparison.
>
> > **Question 5**: For the text description below equation (11), why blurring corruption and block noise corruption can be considered independently, i.e., why can we just replace $\epsilon$ with $\tilde{\epsilon}$? Is there any formal proof?
>
> **Re:** We replace $\epsilon$ with $\tilde{\epsilon}$ corresponding to the forward process in Eq.8: $x_t={{VD^p_tV^\mathrm{T}}}x+\frac{\sigma}{\sqrt{1+\alpha^2}}(\mathbf{\epsilon}+\alpha\cdot$$\text{Block}$[s]$(\mathbf{\epsilon'}))$. Since the mixing ratio is fixed, it retains linear properties and can directly replace Gaussian noise in the sampler. This adjustment better bridges the gap between multiple resolutions, achieving improved results.
>
> The blurring corruption process and the block noise corruption process evolve separately depending on the time $t$. The blurring process interpolates between an upsampled low-resolution start and a high-resolution end. The noise corruption otherwise performs a standard diffusion process. These two processes are orthogonal. We have also provided a detailed proof of the sampler deduction for further verification of the sampler in Appendix A.3.
>
> > **Question 6:** For Algorithm 1 in A.4, it seems that the sampling algorithm does not contain the process of the low-frequency diffusion model. Please clarify.
>
> Thanks for your attentiveness. This is a writing error and actually, the original first line in alg1 does not match the actual practice, which is supposed to be “sample $X_N$ ~ low-resolution images”, we have corrected it in our paper. The Algorithm Section is now presented as follows:
>
> https://i.postimg.cc/KcBwRw0T/rdm-sampler.jpg
>
> For the first stage of RDM, we use a standard diffusion sampler borrowed from EDM[3] as follows:
>
> https://i.postimg.cc/4yjLMN3w/edm-sampler.jpg
>
> > **Question 7:** Some minor errors.
> >
> > Point 2: For equation (6), the denominator should be $s$ or $s^2$?
>
> **Re:** Thanks for your careful attention! While in point 2, the denominator is exactly $s$ to keep the variance consistent. Except for this, we have fixed other errors and a second check on writing has been conducted.
>
>
>
> Except for the above contents, we have also conducted further experiments at higher resolutions (512 and 1024) and achieved **SoTA performance on CelebA-HQ 512 and 1024 generation**. This successfully demonstrated RDM's capability in high-resolution image generation. We have added it to paper in Appendix C.1. The results are presented as follows:
>
> https://i.postimg.cc/X71hwLST/celebahq-hres-results.jpg
>
> If our answers above solve your concerns, would you please consider revising your rating upwards?
>
>
> [1] Ramesh A, Dhariwal P, Nichol A, et al. Hierarchical text-conditional image generation with clip latents[J]. arXiv preprint arXiv:2204.06125, 2022, 1(2): 3.
>
> [2] Saharia, Chitwan, et al. "Photorealistic text-to-image diffusion models with deep language understanding." Advances in Neural Information Processing Systems 35 (2022): 36479-36494.
>
> [3] Karras, Tero, et al. "Elucidating the design space of diffusion-based generative models." Advances in Neural Information Processing Systems 35 (2022): 26565-26577.

---

### Meta-Review · Area_Chair_RUgu · 2023-12-12

**Metareview:**

The paper proposes a diffusion framework for medium-resolution, 256*256, image generation. It builds upon EDM and cascaded diffusion model; the key idea is the introduction of locally correlated noise (by the analysis of SNR in frequency domain) to overcome the current cascaded models' drawback. The proposed method outperforms other methods on FID and sFID on face dataset and general class dataset.
Strengths: (1) a new SOTA on 256*256 image generation, (2) analysis of SNRs across frequencies is inspiring, (3) the new idea of the block noise.
Weaknesses: (1) high resolution or medium resolution? The resolution that the paper uses is 256. It lacks support from higher resolutions, like 512 and 1024. (2) overclaiming some of the results, e.g., convergence speed comparison might be unfair, reported training costs might be misleading and SotA FID claims are not entirely correct.
The camera-ready version should address the weakness parts, and the discussions over the other issues that were raised by the reviewers and addressed during the author-reviewer discussion phase.

**Justification For Why Not Higher Score:**

Some overclaims (like high resolution, convergence speed, training speed, FID) should be addressed.

**Justification For Why Not Lower Score:**

Based on the paper's quality and all the reviews and the confidence values, AC felt that this is a solid work despite some flaws and it's worth the promotion.

---

### Decision · Program_Chairs · 2024-01-16

Accept (spotlight)